

# The canopy interception–landslide initiation conundrum: insight from a tropical secondary forest in northern Thailand

Roy C. Sidle[1]

Sustainability Research Centre

5   University of the Sunshine Coast

90 Sippy Downs Drive, Sippy Downs

Queensland 4556, Australia

Email: rsidle@usc.edu.au

10   Alan D. Ziegler

Department of Geography

National University of Singapore

AS2, #03-01, 1 Arts Link, Kent Vale

Singapore 117570

15   Email: adz@nus.edu.au

---

[1] Corresponding author





**Abstract.** The interception and smoothing effect of forest canopies on pulses of incident rainfall and its delivery to the soil has been suggested as a factor in moderating peak pore water pressure development within soil mantles, thus reducing the risk of shallow landslides. Here we provide three years of rainfall and throughfall data in a tropical secondary dipterocarp forest characterized by few large trees in northern Thailand, along with selected soil moisture dynamics, to address this issue. Throughout the sampling period, throughfall was an estimated 88% of rainfall, varying from 86-90% in individual years. Data from 167 events demonstrate that canopy interception was only weakly associated (via a non-linear relationship) with total event rainfall, but not significantly correlated with duration, mean intensity, or antecedent 2-day precipitation (API$_2$). Mean interception during small events ($\leq$ 35 mm) was 17% (n = 135 events) compared with only 7% for large events (> 35 mm; n = 32). Examining small temporal intervals within the largest and highest intensity events that would potentially trigger landslides revealed complex patterns of interception. The tropical forest canopy had little or no smoothing effect on incident rainfall during the largest events. During events with high wind speeds and/or moderate-to-high pre-event wetting, measured throughfall was occasionally higher than rainfall during large event peaks, demonstrating limited buffering. However, in events with little wetting and low-to-moderate wind speed, early event rainfall peaks were buffered by the canopy. As rainfall continued during most large events there was little difference between rainfall and throughfall depths. Comparing both rainfall and throughfall depths to mean intensity–duration thresholds for landslide initiation, throughfall exceeded the threshold in 75% of the events in which rainfall exceeded the threshold for both wet and dry conditions. Throughfall intensity for all the 11 largest events (rainfall = 65-116 mm) plotted near or above the intensity-duration threshold for landslide initiation during wet conditions; five of the events were near or above the threshold for dry conditions. Soil moisture responses during a range of rainfall conditions in large events were heavily and progressively buffered at depths of 1 to 2 m, indicating that the time-scale of short-term smoothing of peak rainfall inputs (i.e., $\leq$ 1 h) has little or no effect on peak pore water pressure at depths where landslides would initiate. Given these findings, we conclude that canopy interception would have little effect on mitigating shallow landslide initiation during the types of monsoon rainfall conditions in this and similar tropical secondary forest sites.

**Key Words:** throughfall, rainfall, canopy interception, soil moisture, shallow landslides, intensity-duration thresholds, tropical secondary forest, Thailand



# 1 Introduction

Mechanisms of slope failure in relatively shallow soil mantles during rain events are generally well understood. Typically a positive pore water pressure develops just above a hydrologic discontinuity in the regolith causing an abrupt decline in shear strength and resultant rapid landslide (e.g., Sidle and Swanston, 1982; Harp et al., 1990; Fernandes et al., 1994; Kuriakose et al., 2008). Alternatively, landslides have been known to occur due to an increase in soil weight and reduction in soil suction as soils wet during events (Sasaki et al., 2000; Lacerda, 2007; Godt et al., 2009; Yamao et al., 2016). In contrast, some interactions amongst vegetation, site hydrology, and slope stability are not as well understood. In particular, the role of canopy interception of precipitation has drawn considerable speculation with little supporting data.

Root systems of woody vegetation contribute significantly to the reinforcement of potentially unstable slopes, especially when roots anchor into stable substrate (Gray and Megahan, 1981; Riestenberg and Sovonick-Dunford, 1983; Roering et al., 2003; Stokes et al., 2009), or enhance lateral reinforcement within the soil mantle (O'Loughlin and Ziemer, 1982; Schmidt et al., 2001; Schwartz et al., 2012). When trees are cut, the root strength declines with time (Sidle, 1991). If forests then regenerate, the period of significantly reduced root strength ranges from about 3 to 20 years after harvesting (Ziemer, 1981; Sidle and Wu, 1999; Imaizumi et al., 2008). However, if forest sites are converted to weak, shallow-rooted agricultural species or plantations, root strength remains low indefinitely (Sidle et al., 2006; DeGraff et al., 2012). While this mechanical reinforcement of shallow soil mantles by roots is well recognized and has been successfully modelled (Sidle, 1992; Wu and Sidle, 1995; Dhakal and Sidle, 2003; Schwartz et al., 2013), the effects of the presence or absence of trees on the hydrological processes of transpiration, interception, water redistribution, and subsequent pore pressure formation in the subsurface remain a topic of controversy, especially when related to shallow landslide initiation (Keim and Skaugset, 2003; Reid and Lewis, 2009; Ghestem et al., 2011; Greco et al., 2013; Dhakal and Sullivan, 2014).

Deep-rooted woody vegetation extracts soil water near potential failure planes during periods of high transpiration; however, such effects typically are not expected to augment slope stability during extended rainy periods when soils are already at field capacity, especially in temperate regions (Megahan, 1983; Sidle and Ochiai, 2006). In the tropics, where evapotranspiration rates are sustained year-round, the potential for modification of the soil moisture regime when trees are removed may be greater. Nevertheless, simulations of soil moisture in a Peninsular Malaysia rain forest indicate that evapotranspiration more significantly affects soil moisture during events preceded by dry conditions than events preceded by wet conditions; it is during these wet periods that shallow landslides are more likely to occur (Sidle, 2005; Sidle et al., 2006).





Because forests intercept and evaporate rain water back to the atmosphere, less rainfall typically reaches the forest floor when canopies are intact (e.g., Rowe et al., 1999; Crockford and Richardson, 2000; Reid and Lewis, 2009; Ziegler et al., 2009; Kato et al., 2013). However, the effects of canopy interception are complicated by antecedent precipitation, wind, rainfall intensity and duration, and canopy structure (Xiao et al., 2000; Scott et al., 2003; Pypker et al., 2005;

Germer et al., 2006; Kato et al., 2013). As such, canopy interception can vary greatly from event to event at a given site (Keim et al., 2004; Ziegler et al., 2009). Most studies show that the percentage of rainfall intercepted by tree canopies is most variable and highest for small events compared to larger events (Filoso et al., 1999; Keim et al., 2004; Germer et al., 2006; Reid and Lewis, 2009; Ziegler et al., 2009; Dhakal and Sullivan, 2014). In addition to interception of rain water, forest canopies have been reported to exert a buffering effect on short-term pulses of incident rainfall (Xiao et

al., 2000; Keim and Skaugset, 2003; Keim et al., 2006). Using a stochastic representation of rainfall, canopy evaporation, and rainfall transfer through the canopy, Keim et al. (2004) showed that effective rain intensity during large events was reduced more for short duration events than for long duration events; during small events such differences with storm duration were not apparent.

Based on reports of interception and intensity smoothing in forest canopies, it has been advocated that canopy

removal could lead to more intense pulses of rainfall infiltrating into forest soils and subsequently higher pore water pressures in the subsurface that could exacerbate landsliding (Rowe et al., 1999; Keim and Skaugset, 2003; Keim et al., 2004; Reid and Lewis, 2007, 2009). If this is true, then in addition to the delayed effect of root decline after tree removal, there could also be an immediate negative effect due to the lack of canopy interception and subsequent potential to increase pore water pressure. While attempts have been made to include canopy interception losses into subsurface

hydrology (Keim et al., 2006) and landslide models (Wilkinson et al., 2002), it is not clear what the mechanistic effects are on slope stability. Most canopy interception studies have been conducted in temperate forests, but this information is especially needed in tropical rain forests to assess the possible effects of canopy removal on landslide initiation in these regions where management pressures are rapidly increasing.

The objectives of this research are three-fold: (1) evaluate rainfall interception by a secondary tropical forest canopy

for a large number of monsoon events; (2) compare throughfall and rainfall rates to intensity–duration threshold relationships established for shallow landslides; and (3) determine the effect of canopy interception on the potential for soil water increases that could trigger landslides. One major question related to slope stability is to determine if canopy interception significantly reduces incident rainfall with respect to established rainfall intensity-duration thresholds for landslide initiation. Another question relates to finding evidence that canopy interception significantly affects event soil

moisture dynamics to an extent that it would influence shallow landslide initiation. These questions are addressed within



the context of a three-year field investigation in a disturbed, secondary hill dipterocarp forest stand in northern Thailand.

## 2 Site description

The Mae ('river' in Thai language) Sa Experimental Catchment, a headwater catchment of the Ping River, is located northwest of Chiang Mai city in northern Thailand (18°54'06.8"; 98°53'14.2"; Figure 1a). The 74.2 km² catchment is mountainous, with elevations ranging from 500 to 1400 m asl. The topography is characterized by steep (some exceeding 45°) slopes and narrow valleys. The geology of the catchment includes granites and gneiss, with some marble and limestone. Soils include Ultisols, Alfisols, and Inceptisols; soil depth typically exceeds 2 m. Land cover is primarily mixed secondary forests and scrublands (together approximately 80% of the land area), with ongoing conversion to intensive agriculture, especially tree crops, floriculture, and greenhouse operations (Figure 1b). These agricultural activities, in addition to ecotourism, support the economies of several small villages. Much of the development, including the building and maintenance of major roads, is located immediately adjacent to the Sa River and its tributaries. A few recent landslides have occurred within the upper catchment (personal observations); most are associated with road runoff. Given the rapid revegetation of these tropical sites, it is difficult to detect older landslides.

The catchment is the site of ongoing investigations of hydrological and land-use change (Sidle and Ziegler, 2010; Bannwarth et al., 2014a,b; Ziegler et al., 2014a,b). Associated instrumentation includes 11 spatially-distributed rain gages and one stream gaging station that monitors discharge and turbidity at the mouth of the catchment at a sub-hourly time scale (Figure 1). Mean annual rainfall in the catchment varies from 1500 to 2000 mm y$^{-1}$. The vast majority of the annual precipitation is delivered as intense rainfall (often exceeding 20 mm h$^{-1}$) during the monsoon season between May and November. The catchment has a mean annual runoff ratio of approximately 30% (Ziegler et al., 2014b).

The throughfall experiment was conducted at station 429 (Figure 1b), which consists of a hydrometeorological tower that measures water and energy fluxes within an upland, dipterocarp forest. Trees are typically 10-16 m tall in the forest; tree trunk diameter ranges from 2-77 cm. Tree density in the plot is moderate: 127 trees within a 350 m² plot. Leaf Area Index ranges from 1.8-3.2, as determined at 117 point locations using fisheye digital photographs and Deta-T Hemiview software (www.delta-t.co.uk). The forest is secondary and is typically burned annually, but fire only affects the understory vegetation.

A total of 34 tree species are found in the plot, along with three types of bamboo: *Bambusa tulda* Roxb (Poaceae); *Bambusa pallida* (Poaceae); and *Dendrocalamus nudus* Pilg. (Poaceae). Dominant tree species are *Shorea obtusa* Wall. Ex Bl. (Dipterecarpaceae) and Quercus kerrii Craib var. kerrii (Fagaceae). Other important species include: *Lithocarpus*





*polystachyus* (Wall. Ex A. DC.) Rehder. (Fagaceae); *Tectona grandis* L. f. (Verbenaceae); *Craibiodendron stellatum* (Pierre) W.W. Sm. (Ericaceae); *Cratoxylum formosum* (Jack) Dyer ssp. Pruniflorum (Kurz) Gogel. (Guttiferae, Hypericeae); *Dipterocarpus tuberculatus* Roxb. Var. tuberculatus (Dipterocarpaceae); *Gardenia sootepensis* Hutch. (Rubiaceae); *Pterocarpus marocarpus* Kurz (Leguminosae, Papilionoideae); *Shorea siamensis* Miq. var. siamensis (Dipterocarpaceae); and *Wendlandia tinctoria* (Roxb.) DC. subsp. Orientalis Cowan (Rubiaceae).

The soil at the site is an Ultisol with a thin (< 20 cm) brown A horizon underlain by a dark red B horizon that extends below a depth of two meters. Saturated hydraulic conductivity declines exponentially from the surface (~136 mm h$^{-1}$) to approximately <5 mm h$^{-1}$ at a depth of 25 cm (n = 3 measurements; unpublished data). Bulk density does not change much over this depth range (1.17-1.38 g cm$^{-3}$). The decrease in saturated hydraulic conductivity is typical of that in other profiles found in southeast Asia (cf. Ziegler et al., 2004; 2006). Macropores and fissures, features that could influence preferential flow through the soil, were not abundant in the subsoil.

## 3 Methods

### 3.1 Measurements

To assess real-time rainfall interception in the forest stand, a tipping-bucket rain gage was mounted on a meteorological tower at a height of 18 m, about 1 m above the tallest canopy trees (820 m asl), to measure incident rainfall (stations 429, Figures 1b). For our analyses we examined all events that occurred during the period from 6 May 2005 to 21 November 2007. To be considered an event, total rainfall during a period had to be ≥ 8 mm with no precipitation break > 4 h occurring.

Soil moisture was monitored at the soil surface and depths of 1 and 2 m in the same forest patch using Campbell Scientific (Logan UT, USA) CS-615 soil moisture probes, connected to a Campbell CR23x data logger. The probes were situated less than 10 m from the throughfall collection system under the forest canopy. Soil moisture measurements were recorded at 20 min intervals (note: these are instantaneous measurements, not means). Water content reflectometer values recorded with the CS-615 were converted to volumetric water contents via sensor-specific calibration curves determined from manual samples collected within the soil profile at the time of installation and during subsequent periods of both wet and dry seasons. During the latter periods, manual samples were collected by augering holes to a 2 m depth near the probe site. Volumetric samples were collected with an AMS bulk density sampler. The calibration curve was determined via linear regression from the reflectometer (independent variable) and paired volumetric water content (dependent variable) data. Details of this calculation are provided by the manufacturer (https://s.campbellsci.com/documents/us/manuals/cs616.pdf).





Throughfall was collected in a system consisting of six 4-m long gutters radiating from a central tipping-bucket device that was installed under the canopy and secured at heights of 0.5-1.0 m above the ground at a slight angle (≤ 6°) to promote rapid drainage (Figure 2a); the angle was based on prior experience (Ziegler et al., 2009). Each collection gutter of the throughfall system was 43.5 mm wide, with a triangular-shaped channel and 25 mm vertical risers to reduce

rain splash loss. All gutters drained into a large tipping-bucket to measure real-time throughfall response for comparison with incident rainfall. The volume of throughfall required to produce one tip was 230-240 cm³ (0.22-0.23 mm). A dynamic calibration correction was then applied to account for differences in tip volume over the range of observed tipping rates (Ziegler et al., 2009).

The volume of rainfall within a given time interval was divided by the total surface collection area of the entire gutter

system, corrected for the angle of inclination (area = 1.044 m²), to calculate total throughfall for the interval. A correction was also applied to account for splash error occurring during high-intensity throughfall. Based on data collected from seven events using a paired tipping bucket rain gage and a throughfall system installed in an open area (i.e., both were used to measure rainfall), we observed that substantial splash loss occurred during high-intensity events. Total event rainfall depths between the devices could be achieved when calibrated tip volumes were increased 50-78%

(via linear regression) during high-intensity periods of events (i.e., for rates of 5-12 tips per minute). We used this relationship to adjust the throughfall rates for high-intensity throughfall measured in this study.

We recognize several limitations in this correction: (1) the correction is based on limited data (unpublished); (2) some splash error probably also occurred at lower intensities (< 5 tips per minute), but the data set does not allow us to quantify it; and (3) the splash error associated with open rainfall may not be the same as that for throughfall, owing

to different drop sizes and drop direction (both of which vary from event to event). Nevertheless, after applying this crude correction, the total event throughfall depths were within the ranges (relative to rainfall) expected for the range of events measured (i.e., the $C_i$ of increasingly large events approached a value of 0; see Figure 3a). Thus, we believe any residual errors due to splash (after correction) are minor; importantly, these errors would not change our final interpretations.

The time interval used to assess both rainfall and throughfall inputs via the respective tipping bucket devices was 1 min. Although the collector was kept in the same location during the three year study, it has an advantage over using conventional, movable, tipping-bucket rain gauges because it integrates throughfall response under much of the variable canopy structure (Ziegler et al., 2009). Spatial integration of throughfall is especially important in tropical forests where multi-tiered canopies create considerable variability in throughfall (Lloyd and Marques, 1988; Dykes, 1997; Konishi et

al., 2006).





### 3.2 Calculations

Canopy interception ratio ($C_i$) is calculated as

$$C_i = (RF - TH)\ RF^{-1} \qquad\qquad [1]$$

where, RF is the incident rainfall during an event or a portion of the event (mm) and TH is the throughfall (mm) during the same period of time. Values of $C_i$ approaching zero indicate no canopy interception for that event or period of the event.

As a conservative predictive measure for shallow landslides, a number of regional studies have generated rainfall intensity–duration relationships (e.g., Larson and Simon, 1993; Aleotti, 2004; Guzzetti et al., 2007; Dahal and Hasegawa, 2008) based on an earlier global concept developed by Caine (1980). Sidle and Ochiai (2006) modified Caine's global intensity–duration threshold by removing some very short and very long (> 10 days) events that misrepresented rainfall – landslide initiation data. The resulting relationship is given as:

$$I = 13.58\ D^{-0.38} \qquad\qquad [2]$$

where I is the mean event intensity (mm h$^{-1}$) and D is the duration of the event (h). To assess the effects of antecedent rainfall on the intensity–duration relationship, all of Caine's (1980) data that included 2-day antecedent rainfall (API$_2$) together with new data were plotted separately for API$_2$ ≤ 20 mm and API$_2$ > 20 mm (Sidle and Ochiai, 2006). Two-day antecedent rainfall was used because this parameter correlated well with maximum piezometric response in unstable hollows (Sidle, 1992). The intensity–duration relationship developed for the events preceded by dry (≤ 20 mm) antecedent conditions is the following (Sidle and Ochiai, 2006):

$$I = 19.99\ D^{-0.38} \qquad\qquad [3]$$

where I and D are as in Eq. 2. For the events preceded by wet antecedent conditions, the following relationship is used:

$$I = 12.64\ D^{-0.49} \qquad\qquad [4]$$

where I and D are as above. The global modified Caine threshold (Eq. 2) is lower than the dry condition threshold (Eq. 3) for all combinations of event intensity–duration. Furthermore, because the equations are based on duration, the modified Caine threshold is higher than the wet condition threshold (Eq. 4) for events exceeding 1-h duration.





## 4 Results

### 4.1 Canopy interception for all events

The maximum, minimum, and mean rainfall totals for the 167 recorded events during the three year study were 116.4, 8.1, and 24.3 mm, respectively. The corresponding totals for throughfall were 114.6, 5.7, and 21.3 mm, respectively.

Event mean rainfall intensity (depth/duration) ranged from 0.5 to 88.0 mm h$^{-1}$. The mean event intensity of all 167 events was 9.1 mm h$^{-1}$. The duration of the 167 events ranged from about 9 min to 57 h, with a mean duration of about 7 h.

A total of 52, 59, and 56 events where monitored in 2005, 2006, and 2007 (Table 1). Mean event size ranged from 22-28 mm. Rainfall depths for the three years varied from 1149-1678 mm; the corresponding throughfall depth range was 1037-1450 mm (Table 1). Annual estimates of throughfall (fraction of rainfall) were 0.90, 0.86, and 0.87, respectively. We can only speculate that annual variations result from minor changes in canopy characteristics (based on observations) and differences in event rainfall characteristics. Further, we believe the inherent error in the calculation of throughfall for any one event is on the order of ± 6%. The differences in the yearly calculations are within this tolerance.

The three-year throughfall estimate was 0.88, which is near the higher end of values reported for forests in Southeast Asia (e.g., Sinun et al., 1992; Dykes, 1997; Konishi et al., 2006; Ziegler et al., 2009; Tanaka et al., 2015). The throughfall estimate may be slightly elevated because of the following reasons: (a) the stand was a recovering secondary forest (i.e., lacking a multi-story canopy) with low LAI (1.8-3.2); (b) the event-based estimate does not include many very small events when canopy interception is expected to be high (i.e., we only report throughfall for events representing 78, 87, and 82% of incident rainfall entering the forest in 2005, 2006, and 2007; data not shown); and (c) under-catch of rainfall above the canopy during windy conditions. Nevertheless, we believe the method provided reasonably accurate data for this type of analysis.

For the 167 events, throughfall ranged widely from 62 to 129% of rainfall (Figure 4). When expressed as a canopy interception ratio ($C_i$, Eq. 1), values ranged from -0.29 to 0.38, with the mean for all 167 events being 0.15 (Figure 3). Throughfall was greater than incident rainfall during 9 events (Figure 4b), but only for four events was the difference greater than 6%, a value we consider to be approximately the uncertainty in the throughfall estimate. Most of the events where throughfall was greater than rainfall occurred in 2007 (n = 4), with one and three occurring in 2005 and 2006, respectively (Figure 4b). We consider the four exceptionally low values (ranging from -0.18 to -0.24) to be outliers in





this analysis, as we cannot fully explain the nature of the very high throughfall that generated them (see discussion that follows). Both the boundary of uncertainty and the outliers are shown in Figure 4b.

A total of 35 events had $C_i$ values ≥ 0.25 (Figure 3a). Most of these events were small-to-moderate in size (range = 8.7-32.0 mm; median = 12 mm), but rainfall depth in three events exceeded 30 mm. These 35 events ranged from 0.5 to 7.4 h in duration; event intensities ranged from 2 to 37 mm h$^{-1}$. Three events had mean intensities > 20 mm h$^{-1}$. The events also ranged greatly in antecedent moisture conditions: API$_2$ ranged from 0.0 to 60 mm.

Except for four events with anomalously low $C_i$ values, the overall tendency was for $C_i$ to converge towards zero as total rainfall increased, particularly beyond 60 mm (Figure 3a). However, the relationship between $C_i$ and total rainfall was not strong (non-linear regression; $R^2$ = 0.11; significant at α = 0.05; Figure 3a). No meaningful relationship existed between $C_i$ and API$_2$, event duration, or mean event intensity (Figures 3b-d). However, during longer events with short periods of high intensity, mean intensity would not be a good index to compare with $C_i$.

## 4.2 Canopy interception during large storms

Because many shallow landslides occur after a high-intensity burst of rainfall that follows an initial period of lower intensity rain (e.g., Okuda et al., 1979; Sidle and Swanston, 1982; Sidle and Chigira, 2004), we focus mainly on the largest and longest events (Table 2). Eleven of the events summarized in Table 2 have total rainfall depths > 65 mm; and one had a duration > 55 h (event #1; total rainfall = 40 mm). The eleven large events have low $C_i$ values (-0.01 to 0.11), indicating that most rainfall was converted into throughfall (open circles in Figure 3). The long-duration event #1 had a relatively high $C_i$ value (0.14; triangle in Figure 3b).

Events #54 and #158 had $C_i$ values ≤ 0 (Table 2). Event #54 was short-duration, high intensity with moderately high wind speed, while event #158 had relatively low intensity and wind speed, but high surface soil moisture prior to the storm. The high intensity and moderately high wind speed during event #54 likely generated non-vertical rainfall that may have been underestimated in the gauge above the canopy and may have dislodged and transferred rainfall from proximate trees to the plot canopies. The slightly negative $C_i$ (-0.01) during event #158 was likely affected by wet conditions (soil moisture at the surface and 1 m depths were 0.45 and 0.42 g cm$^{-3}$, respectively) at the onset of the storm.

Five events had canopy interception values ranging from 0.02-0.05 (#99, #156, #48, #56, #63; Table 2), indicating very limited canopy buffering during these large (65-116 mm) events. Durations and intensities of these five events were variable, ranging from 2.4 to 38.8 h and 5.3-41.6 mm h$^{-1}$, respectively. High antecedent precipitation (API$_2$ = 11-22 mm) and maximum wind speed (2.3-4.2 m s$^{-1}$) occurred prior to and during four of these events, while event #63 had the driest antecedent conditions (Table 2).



The large event with greatest interception (#1; $C_i$ = 0.14) had the driest antecedent conditions ($API_2$ = 0 mm) of all large events considered and very low maximum wind speed (1.9 m s$^{-1}$). Because of these conditions and the relatively small total rainfall depth (40 mm), canopy interception was likely higher than that for the other large events (Figure 3b). We included event #1 as a "large" event because of its exceptionally long duration (> 56 h).

The remaining four large events with intermediate $C_i$ values ranging from 0.09-0.11 (#115, #96, #85, and #26) all had relatively long durations (10.6-29.7 h), moderate intensities (2.4-7.4 mm h$^{-1}$), and relatively low maximum wind speeds (2.2-2.7 m s$^{-1}$) (Table 2). The main difference among these events was the much higher $API_2$ value for event #96 (52 mm) compared to the other three events (0-1 mm).

Collectively, the variation in rainfall characteristics (e.g., duration, intensity) and other hydro-climatic phenomena
(e.g., maximum wind speed, $API_2$) demonstrate the inherent variability in (or the measurement of) canopy interception across a range of large events. In most cases is it difficult to pinpoint the key factor or combination of factors dictating $C_i$ because meteorological conditions (rainfall intensity, wind speed) vary across time and space scales that our methodology does not measure. Again, we consider the error in the interception estimate to be on the order of 6%.

**4.3 Throughfall and rainfall patterns during large storms**

We also examined the pattern of incident rainfall versus throughfall during the six of the largest events (Figure 5). This sub-group includes the three events (#54, #56, and #156) with the highest intensities (36.9-45.6 mm h$^{-1}$), short durations (1.8-2.8 h), and low $C_i$ values (0.0-0.3). Three other events (#63, #99, and #158) had low intensities (2.0-6.8
20   mm h$^{-1}$), long durations (16.4-38.8 h), and a range of $C_i$ values (-0.01 to 0.05). For events #56, #99, and #156, throughfall exceeded rainfall during early large peaks, but was generally lower than rain intensity during latter parts of these events (Figures 5a,b,c). Short-duration event #54 was characterized by an initial burst of rainfall intensity of nearly 3 mm min$^{-1}$ (Figure 5d), which immediately translated into substantial throughfall despite the initially dry canopy ($API_2$ = 0; $SM_{0m}$ = 0.17; Table 2). Event #158 produced a complicated pattern of throughfall response – during the initial
rain burst, throughfall exceed rainfall, but after about a 5-h period of little precipitation, rainfall exceeded throughfall for the rest of the event (Figure 5f). Throughout most of event #63, rainfall exceeded throughfall (Figure 5e). The three largest events (#56, #99, and #156) exhibited minor canopy storage during early, low intensity rainfall; however, during large subsequent peaks (> 100 mm h$^{-1}$, minutely rates) throughfall exceeded rainfall (Figures 5a,b,c). Moderate to high maximum wind speeds occurred during these three events (2.9-4.2 m s$^{-1}$). The highest wind speed was associated with
the largest rainfall event #56, during which throughfall depth was similar to rainfall depth (115-116 mm; $C_i$ = 0.02). In addition, wet canopy conditions preceding these three events are supported by high $API_2$ (11-22 mm) and associated surface soil moisture (0.40-0.48 g cm$^3$) values. During the early peaks of large events with relatively low intensity and





wind speed (#63 and #158), more rain water was likely stored in the canopy (Figure 5e,f). The findings of early storage agree with the generally accepted idea that canopies store a larger proportion of rainwater during the early stage of events (e.g., Xiao et al., 2000; Zeng et al., 2000; Iida et al., 2012).

## 4.4 Soil moisture dynamics

We examined the soil moisture dynamics near the soil surface and at depths of 1 m and 2 m under the forest canopy to ascertain the effects of canopy buffering on water movement in the soil to depths where shallow landslides may occur. In particular, we focus on periods during events when the canopy exerts a maximum influence on short-term incident rainfall. Such canopy interception effects have been suggested to provide benefits to slope stability during large, landslide-producing events (Rowe et al., 1999; Keim and Skaugset, 2003; Reid and Lewis, 2009). We show soil moisture dynamics for the same six events used to assess intra-storm patterns of rainfall/throughfall (Figure 6), but also consider changes in the other large events (Table 2). The six events include a range of dry and wet antecedent moisture conditions (e.g., $API_2$ ranged from 0 to 22 mm). Corresponding surface, 1-m, and 2-m initial soil moisture ranged from 0.17-0.48 $m^3 m^{-3}$, 0.34-0.42 $m^3 m^{-3}$, and 0.33-0.37 $m^3 m^{-3}$, respectively (Table 2; Figure 6). These values indicate large differences in pre-event soil moisture near the soil surface, but not at depths where landslides may initiate.

Events #56, #99, #156, and #158 are representative of relatively wet antecedent conditions. Initial surface soil moisture values for these events range from 0.40-0.48 (Figure 6). Throughfall infiltration into the soil produced peaks in surface soil moisture that lagged behind throughfall peaks by typically 20-60 min (Figure 6; note that soil moisture is measured every 20-min). In some cases, surface soil moisture increased > 0.1 $m^3 m^{-3}$ during the event (e.g., events #56, #156, #158). The rainfall rates triggering these increases typically exceeded 60-100 mm $h^{-1}$ for 20 min periods (Figure 6a,c,f). During event #99, two periods of rainfall resulted in corresponding peaks in surface soil moisture. For all four events, increases in soil moisture at the 1-m depth occurred 100-180 min after the onset of rainfall, or 30-90 min following the maximum rainfall and/or throughfall rate (Figures 6a,b,c,f).

During events with drier antecedent conditions (#54 and 63; surface soil moisture ≤ 0.34 $m^3 m^{-3}$), much greater wetting occurred in the surface soil (increases of 0.14-0.27 $m^3 m^{-3}$) compared with that during wetter antecedent conditions (Figures 6d,e). Event #54 was characterized by an initial burst of rainfall on dry soil (0.17 $m^3 m^{-3}$), which rapidly elevated surface soil moisture over the next hour (Figure 6d). Event #63 consisted of two rainfall/throughfall peaks that produced corresponding peaks in surface soil moisture with short lags (20-40 min). This event also caused a lagged increase in soil moisture at the 1-m depth, whereas subsoil moisture during event #54 was unaffected during this shorter (1.8 versus 10.6 h) event.




Soil moisture at the 2-m depth exhibited delayed and very minor increases in response to inputs of rainfall/throughfall during events #156 and #158 only (Figures 6c,f). Both events occurred under some of the wettest conditions observed (surface soil moisture = 0.45-0.48 $m^3 m^{-3}$). Although the events had very different durations (2.4 versus 16.4 h), the increases at 2-m occurred more than an hour after the last rainfall peak, at which time most of the

5 event rainfall had already occurred. It is not clear why the largest event #56 (116 mm falling in 2.8 h on wet soil) did not affect soil moisture at the 2-m depth. Three other large events summarized in Table 2 produced soil moisture changes at 2 m (#96, #85, and #48). In all cases, the maximum observed change in soil moisture at 2-m was only on the order of 0.03 $m^3 m^{-3}$.

All events that increased soil moisture at the 2-m depth had saturated to nearly saturated surface soils (surface soil

moisture values ≥ 0.45 $m^3 m^{-3}$; Table 2; Wetting$_{max}$). Together with these wet surface conditions, total depth of throughfall (or rainfall) appeared to be more important than event intensity in propagating water fluxes to the 2 m depth. In contrast, the events that only affected surface soil moisture had initial soil moisture values ≤ 0.34 $m^3 m^{-3}$ (API$_2$ = 0.0 for all three events; Table 2). Throughfall depths during these drier antecedent conditions ranged from 34-81 mm. Event #54 had the highest measured event intensity (45.6 mm $h^{-1}$), but produced no increases in subsoil

moisture. Intermediate of these responses, some events with initial soil moisture conditions ranging from 0.30-0.45 $m^3$ $m^{-3}$ produced soil moisture increases down to one meter depth for a variety of throughfall inputs (64-115 mm) occurring over 2.8 to 18.9 h (Table 2). Observed changes in soil moisture at 1 m were on the order of 0.02-0.08 $m^3 m^{-3}$. Even during the largest events, soil moisture contents were well below saturation at the 1 m and 2 m depths.

### 4.5 Rainfall duration–intensity landslide thresholds

To further assess the potential of the monitored events to initiate shallow landslides, we compared incident rainfall and throughfall to three intensity-duration landslide threshold relationships (Eqs. 2, 3, and 4). Considering all 167 events, regardless of antecedent rainfall, 37 rainfall events exceeded the threshold (Eq. 2) for potential landslide initiation, while throughfall from 30 events fell on or above this threshold (Figure 7a). This difference of seven events is associated with 10 events with rainfall > throughfall (positive canopy interceptions ranging from 0.03 to 0.34) and three events with

throughfall > rainfall (negative $C_i$ ranging from -0.06 to -0.24). Five of these events had rainfall depths > 25 mm and six had intensities > 10 mm $h^{-1}$ (not shown). Only one had an API$_2$ value > 20 mm – event #80, 29 mm during a period of 3.3 h (9 mm $h^{-1}$) with a $C_i$ = 0.14 (not shown).

We segregated the 167 events into those preceded by dry and wet conditions to compare with Eqs. 3 and 4 (Figure 7b,c). A total of 120 events were preceded by dry conditions (API$_2$ ≤ 20 mm); 47 events were preceded by wet conditions

(API$_2$ > 20 mm). For dry conditions, 16 and 12 rainfall and throughfall events, respectively, plotted above the



corresponding threshold (Eq. 3). For the events where both rainfall and throughfall were above the threshold, durations did not exceed three hours and intensities were greater than 26.7 mm h$^{-1}$ (Figure 7b). Events where rainfall exceeded the threshold but throughfall did not were characterized by low to moderate rainfall depths (12-30 mm), appreciable antecedent rainfall (API$_2$ = 7-18 mm; except for one event with 0 mm), positive $C_i$ values (0.07-0.38), and short

durations (0.3-1.7 h). In contrast, 12 and 9 rainfall and throughfall events, respectively, plotted above the threshold for wet conditions (Eq. 4; Figure 7c). The nine events with throughfall above the threshold were variable in length (0.42-11.8 h) and intensity (4.7-54.1 mm h$^{-1}$). The three events in which incident rainfall exceeded the threshold but throughfall did not were similar in canopy interception ($C_i$ = 0.20-0.23); duration (3.4-4.6 h); and event intensity (7.1-7.7 mm h$^{-1}$). Rainfall depth for these events varied from 24-34 mm.

Nearly all of the 12 largest events (Table 2) plot on or above the threshold for wet conditions (Eq. 4; Figure 7d). The long-duration event #1 (56.6 h) plotted well below all thresholds (Figure 7d). Throughfall for three events (#54, #56, and #156) plotted well above the threshold for dry conditions demonstrating their potential for landslide generation, despite having relatively short durations (about 2-3 h) (Table 2; Figure 7d). Collectively, these comparisons show the limited potential of canopy interception to reduce the probability of landslide initiation during large annual

storms (e.g., those listed in Table 2), particularly under wet antecedent conditions.

## 5. Discussion

Our estimated 3-year interception loss from a tropical secondary forest in northern Thailand based on 167 storms with total precipitation ≥ 8 mm was 12%. Interception losses for all individual events ranged from -29% to 38% and from

20    -1% to 11% during the 11 largest storms. Overall, we found that events with larger total precipitation had lower rates of interception compared to smaller events (albeit weakly correlated), which agrees with most other studies (e.g., Filoso et al., 1999; Keim et al., 2004; Germer et al., 2006; Reid and Lewis, 2009; Bäse et al., 2012). Given the spatially distributed gutter system we employed to collect throughfall under this secondary forest stand, we believe our estimates are realistic within a measurement error of about 6%.

The measured interception losses in this tropical secondary dipterocarp forest are on the low side of most ranges reported for temperate and semi-arid canopies (e.g., Xiao et al., 2000; Iida et al., 2005; Reid and Lewis, 2007; Kato et al., 2013; Allen et al., 2014; Swaffer et al., 2014; Nanko et al., 2016). However, our values and variabilities are very similar to those reported in an open tropical Brazilian rainforest with many palm trees (10.2±5.6%) where similar magnitudes and numbers of storms were recorded (Germer et al., 2006). Interception losses in other native and

secondary Amazonian forests have been reported in the range from about 6% to 22% (Lloyd and Marques, 1988;





Elsenbeer et al., 1994; Filoso et al., 1999; Tobón Marin et al., 2000; Bäse et al., 2012; Zanchi et al., 2015). Several studies in both native and plantation forests in southeast Asia that experience monsoon storms reported similarly low interception losses in the range of about 7 to 20% (Sinun et al., 1992; Dykes, 1997; Konishi et al., 2006; Ziegler et al., 2009; Tanaka et al., 2015). Given that the preponderance of the research on canopy interception has been conducted

in temperate and arid or semi-arid environments, it is not surprising that this secondary tropical forest in Thailand has relatively lower canopy interception during intense monsoon events.

These slightly lower values of interception we measured may reflect a combination of factors that vary among individual events. Stable isotope differences between throughfall and rainfall during low intensity events show the complexity of pre-event storage on contributions to throughfall in a conifer forest in the Oregon Cascades, but indicate

that the release of residual water stored in canopies may be significant (Allen et al., 2014). Given the wet, humid conditions at the Thailand site, it is possible that pre-event canopy wetness may have augmented throughfall during large events as evidenced by high soil moisture and/or $API_2$. In other large events, it appears that the higher maximum 20-min wind speed was a factor in dislodging water from within plot and surrounding canopies to restrict interception. Many studies have shown that storage of water in tree canopies is reduced under windy conditions (e.g., Hutchings et

al., 1988; Llorens and Gallart, 2000; Xiao et al., 2000; Keim and Skaugset, 2003; Kato et al., 2013). While wind can increase evaporation and dry the canopy during storms (e.g., Kelliher et al., 1992; Xiao et al., 2000), it may increase measured throughfall by increasing canopy drip, changing the angle of incoming rainfall, and capturing wind-blown rain from adjacent trees (Xiao et al., 2000; Ziegler et al., 2009). Non-vertical rainfall, which is common during windy conditions associated with many monsoon storms, is also often under-recorded by small, standard gauges (Kamph and

Burges, 2010). Finally, canopy "drip points" or "pour points" may develop, where intercepted water is channelled preferentially to the collector (Konishi et al., 2006; Ziegler et al., 2009); these points are dynamic and change as canopies develop and environmental conditions change during the storm (e.g., wind) change. Throughfall measurements during four moderate-sized events with anomalously low $C_i$ values (indicated as outliers in Figures 3 and 4) may also have been affected by a combination of these factors.

It should be noted that we did not account for losses due to interception of litter cover on the forest floor, which in some ecosystems can be significant (e.g., Kelliher et al., 1992; Gerrits et al., 2010). Nevertheless, such interception would be little affected by forest removal and subsequent regeneration, unless significant site disturbance occurred during logging. Thus, the ultimate influence of changes in litter interception on pore water pressures at soil depths that could trigger landslides are expected to be minimal (Dhakal and Sullivan, 2014).

We found no evidence of canopy smoothing of incident rainfall on throughfall during large events; such smoothing has been suggested as a factor in moderating peak pore water pressures within soil mantles, thus reducing the risk of





shallow landslides (Rowe et al., 1999; Keim and Skaugset, 2003; Keim et al., 2004; Reid and Lewis, 2009). The throughfall data we present from northern Thailand are unique in terms of number of events and temporal resolution, which allows us to better assess peak responses related to rainfall inputs. During six of the largest events that would potentially trigger landslides, throughfall intensity actually exceeded rainfall intensity during the largest storm peaks in five of the six events (Figure 5). In event #63, which had the lowest peak intensity of these six events, peak rainfall intensity exceeded peak throughfall intensity. Although the peak intensities of rainfall and throughfall differed amongst events, no smoothing (i.e., flattening) of throughfall peaks relative to rainfall peaks was evident. While a few studies have alluded to intensity smoothing by forest canopies on throughfall regime (Xiao et al., 2000; Keim et al., 2004, 2006; Nanko et al., 2016), only one presented specific evidence of intra-storm smoothing (i.e., flattening) of rainfall peaks (Xiao et al., 2000). The only storm that Xiao et al. (2000) presented was small (13 mm) and low-intensity compared to the monsoon events in northern Thailand, and only throughfall under one oak was assessed. In some cases, the smoothing effects are derived through modelling (Keim et al., 2004, 2006), but even in these cases, the effects on pore water pressure in the substrate during events were low.

Furthermore, only soil moisture in the surface horizon was responsive to individual rainfall peaks during the same six large events shown in Figure 5. Soil moisture increases at the depth of 1 m were highly dampened, lagged the rainfall peak by nearly an hour or more, and never approached saturation (Figure 6). At the depth of 2 m, near where shallow landslides would typically initiate in this terrain, only two of these large storms produced very dampened and minor increases in soil moisture; no soil moisture response at 2 m was recorded in the four other large events. This dampened or lack of soil moisture response at depth, shows that even during events with higher peak rainfall versus throughfall inputs (e.g., event #63; Figure 5e), the impact on pore water pressure at the depth of a potential failure plane would be insignificant (Figure 6e).

The comparisons of incident rainfall and throughfall to established intensity-duration landslide threshold relationships allow us to compare the potential for rainfall-initiated landslides in secondary tropical forests (throughfall measurements) versus cutover or converted sites (incident rainfall). Most large events had intensity-duration relationships that fell above global thresholds for potential landslide initiation during wet conditions. The three events that greatly exceeded the most conservative threshold (i.e., for dry conditions) were very short-duration (2-3 h), high intensity (37-46 mm h$^{-1}$) storms. Considering that peak rainfall intensity was generally not much greater than corresponding peak throughfall intensity during large events and that soil moisture response at depths where landslides may initiate ($\geq$ 1 m) did not respond rapidly to peak rainfall inputs, it appears that canopy interception would have little or no influence on mitigating pore water accretion at depths where shallow landslides typically occur in this secondary tropical forest. Furthermore, similar behaviour of incident rainfall and throughfall during individual events with respect



to intensity-duration thresholds for landslide initiation support our conclusion that canopy interception at the site has negligible influence on landslides for the rainfall conditions we observed.

# 6 Conclusions

Our examination of the effects of canopy interception in a secondary dipterocarp forest of northern Thailand on the potential for shallow landslide initiation revealed some interesting findings. Compared to temperate and semi-arid forests, throughfall in our secondary forest plot was relatively high owing to wind effects (transferring rain water from surrounding trees and causing under-catch of precipitation), wet and humid antecedent conditions, and preferential channelling of canopy drip into collection troughs. Nevertheless, our throughfall measurements are in line with many
values reported from both native and secondary forests in Amazonia and elsewhere southeast Asia.

Few studies have reported intra-storm comparisons of incident rainfall and throughfall at temporal resolutions that could be used to assess effects on shallow landslide initiation (i.e. ≤ 1 h). Most studies that have noted smoothing effects of canopy interception on incident rain intensity (that would potentially reduce the risk of landslides) seem to focus on the lowering of throughfall relative to rainfall peaks or how the perceived flattening (smoothing) of throughfall peaks
can be represented in models (Keim and Skaugset, 2003; Keim et al., 2006; Reid and Lewis, 2009; Greco et al., 2013; Nanko et al., 2016). None of these studies conducted across a range of temperate forest cover types show any physical evidence that smoothing of rainfall peaks by forest canopies during large storms lowered soil moisture or pore pressures at depths that would reduce landslide susceptibility. In contrast, the few studies that have presented throughfall versus rainfall inputs at small time intervals for large events, show little or no evidence of flattening of peaks (Iida et al., 2012;
Safeeq and Fares, 2014; Dhakal and Sullivan, 2014). Only the study of Xiao et al. (2000) measured intensity smoothing effects of single oak canopy during a relatively small storm (13 mm). Our results from many large and intense monsoon events in northern Thailand clearly show that these secondary tropical forest canopies have negligible smoothing effects on incident rainfall peaks.

Although numerous studies have noted the reduction of throughfall relative to incident rainfall, there is little
information on effects of the reduction of rainfall under canopies on soil moisture dynamics. Our findings show that soil moisture response is quite dampened or even non-responsive at depths where potential failure planes exist. These data coupled with our analysis of mean rain intensity – duration thresholds that have been used to estimate the lower global limit of rainfall conditions that may trigger shallow landslides, show that throughfall for the 11 largest events exceeded thresholds for wet antecedent conditions. As such, there is little evidence that canopy interception in this
secondary tropical forest has any mitigating effect on shallow landslides. More likely and much better documented



anthropogenic causes of landslide increases in similar tropical environments include root strength deterioration following timber harvesting, forest conversion, or swidden (e.g., Harper, 1993; Sidle et al., 2006; DeGraff et al., 2012), roads and trails (e.g., Douglas et al., 1999; Chappell et al., 2004; Sidle and Ziegler, 2012), and possibly the effects of increased antecedent soil moisture following clearing or conversion of forest cover (Sidle et al., 2006).

*Acknowledgements.* Gratitude is expressed to Aaron Pruitt for his assistance in data analysis. We also acknowledge Tom Giambelluca, Mike Nullet, and Jefferson Fox for their efforts in the establishment of the Mae Sa experiment site.

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




Figure 1: Site map of the Mae Sa experiment site in northern Thailand. Panel (a) shows the catchment location in Thailand, the topography and major stream channels. Panel (b) shows major land covers in the Mae Sa catchment including hillslope and plantation agriculture (AG, 23%), greenhouse agriculture (GH, 7%), urbanized or peri-urban areas (U, 8%), and forest cover with various degrees of disturbance (F, 62%). Grid cell dimensions are 2 x 2 km. Rectangles demarcate hydro-meteorological measurement sites. Streamflow, total suspended solids, particulate organic carbon, and particulate organic nitrogen were measured at the stream gage station 434. Rainfall is measured at all other numbered hydro-meteorological stations (rectangles). The throughfall investigation in this paper was conducted at station 429, where rainfall, throughfall, and soil moisture where monitored.

Figure 2: (a) Stationary collector with six collection troughs (gutters). (b) Schematic of stationary gauge tipping bucket mechanism (located inside the collector base); the inset shows the dimensions for the collection troughs (gutters).

Figure 3. (a) Canopy interception ratio ($C_i$) versus total event rainfall for all monitored events. The non-linear regression curve desribes the tendency for $C_i$ to approach 0 as total rainfall increases. (b) $C_i$ plotted with respect to event duration (h). (c) $C_i$ plotted against 2-day antecedent precipitation ($API_2$). (d) $C_i$ plotted with respect to mean event intensity (mm h$^{-1}$). The open circles and triangle in all panels refer to the 11 large and one long storm investigated in detail (Table 2). The four events that plot below the dashed line are considered outliers.

Figure 4. (a) Comparison of throughfall to runoff depths for the three years of study (2005-2007). (b) Throughfall fraction of rainfall for events ranging in depths of 8 to 116 mm, during the three-year study. Uncertainty of any throughfall measurement is estimated to be ±6% (indicated in the figure for the case of throughfall = rainfall). The four values labelled "outliers" are exceptionally high estimates for which we cannot completely explain the possible errors.

Figure 5. Five-minute running means of throughfall (solid) and rainfall (hashed) for the six largest storms (Table 2).

Figure 6. Volumetric soil moisture response at three depths during the six largest events (Table 2). Events with initial surface soil moisture (0 cm) ≥ 0.40 m$^3$ m$^{-3}$ are considered to have wet antecedent conditions: #56 (116 mm of rainfall); #99 (111 mm); #156 (89 mm); and #158 (78 mm). The other two events, #54 (81 mm) and #63 (77 mm), are associated with drier antecedent moisture conditions (≤ 0.30 m$^3$ m$^{-3}$). All data are plotted as 20-minute aggregated values. Soil moisture curves in all panels correspond to surface (thin line), 1-m (hashed line), and 2-m (thick line) depths (key shown in panel a). The thickness of each rainfall bar has no meaning; each represents a 20-min value.

Figure 7. Comparison of event rainfall (open circles) and throughfall (solid circles) intensity-duration relationships with modified Caine thresholds for shallow landslide initiation. (a) All 167 events are plotted against the general threshold



(Eq. 2). (b) Events associated with dry conditions ($API_2 < 20$mm) are plotted against the threshold defined by Eq. 3. (c) Events associated with wet conditions ($API_2 \geq 20$mm) are plotted against the threshold defined by Eq. 4. (d) The 12 largest storms summarized in Table 2 are plotted against all thresholds (Eqs. 2-4). Some apparently missing throughfall data points plot behind their paired corresponding rainfall value.





**Table 1. Number of events sampled each year and corresponding rainfall and throughfall totals.**

| Year | Events | Mean depth (mm) | Rainfall (mm) | Throughfall (mm) | Throughfall (-) |
|------|--------|-----------------|---------------|------------------|-----------------|
| 2005 | 52 | 22 | 1149 | 1037 | 0.90 |
| 2006 | 59 | 28 | 1678 | 1450 | 0.86 |
| 2007 | 56 | 22 | 1235 | 1078 | 0.87 |
| total | 167 | 24 | 4062 | 3564 | 0.88 |

Mean depth refers to the mean rainfall depth of the 52, 58, or 56 events in a given year (2005, 2006, 2007). Throughfall is listed as both a depth and a fraction of total rainfall.





10   **Table 2. Characteristics of 12 large/long rainfall events considered in this analysis.**

| Event | RF (mm) | TF (mm) | Ci (-) | D (h) | I (mm h⁻¹) | API₂ (mm) | Wind (ms⁻¹) | SM (0m) (m³ m⁻³) | SM (1m) (m³ m⁻³) | SM (2m) (m³ m⁻³) | Wetting$_{max}$ (-) |
|---|---|---|---|---|---|---|---|---|---|---|---|
| 156 | 88.54 | 85.48 | 0.03 | 2.4 | 36.9 | 22 | 2.9 | 0.48 | 0.41 | 0.37 | 2 |
| 85 | 78.04 | 69.43 | 0.11 | 19.3 | 4.1 | 1 | 2.2 | 0.44 | 0.41 | 0.37 | 2 |
| 96 | 70.95 | 63.65 | 0.10 | 29.7 | 2.4 | 52 | 2.7 | 0.46 | 0.41 | 0.37 | 2 |
| 158 | 69.76 | 70.67 | -0.01 | 16.4 | 4.2 | 6 | 1.8 | 0.45 | 0.42 | 0.37 | 2 |
| 48 | 65.02 | 62.81 | 0.03 | 12.2 | 5.3 | 15 | 3.0 | 0.45 | 0.39 | 0.36 | 2 |
| 56 | 116.44 | 114.66 | 0.02 | 2.8 | 41.6 | 11 | 4.2 | 0.40 | 0.34 | 0.33 | 1 |
| 99 | 111.14 | 106.07 | 0.05 | 16.4 | 6.8 | 16 | 3.7 | 0.45 | 0.41 | 0.37 | 1 |
| 115 | 78.64 | 70.10 | 0.11 | 10.6 | 7.4 | 1 | 2.2 | 0.30 | 0.34 | 0.33 | 1 |
| 26 | 70.22 | 63.97 | 0.09 | 18.9 | 3.7 | 0 | 2.2 | 0.34 | 0.40 | 0.37 | 1 |
| 54 | 81.28 | 80.87 | 0.00 | 1.8 | 45.6 | 0 | 3.2 | 0.17 | 0.34 | 0.33 | 0 |
| 63 | 77.07 | 75.54 | 0.02 | 38.8 | 2.0 | 0 | 2.3 | 0.34 | 0.40 | 0.36 | 0 |
| 1 | 39.91 | 34.13 | 0.14 | 56.6 | 0.7 | 0 | 1.9 | 0.29 | 0.34 | 0.33 | 0 |

RF is rainfall; TF, throughfall; $C_i$, canopy interception; D, duration; I, intensity (RF/D), API₂, two-day antecedent precipitation index; Wind, maximum 20-min wind speed (recorded above the canopy); $SM_{xm}$, soil moisture measured at 0-, 1- and 2-m depths; and Wetting$_{max}$, the maximum
15   soil depth (m) at which wetting during the event was measured by the soil moisture probes. The first 11 events are largest recorded (based on depth); event #1 is the longest (ranked only 23rd in size). Events are ranked by Wetting$_{max}$, then rainfall depth.



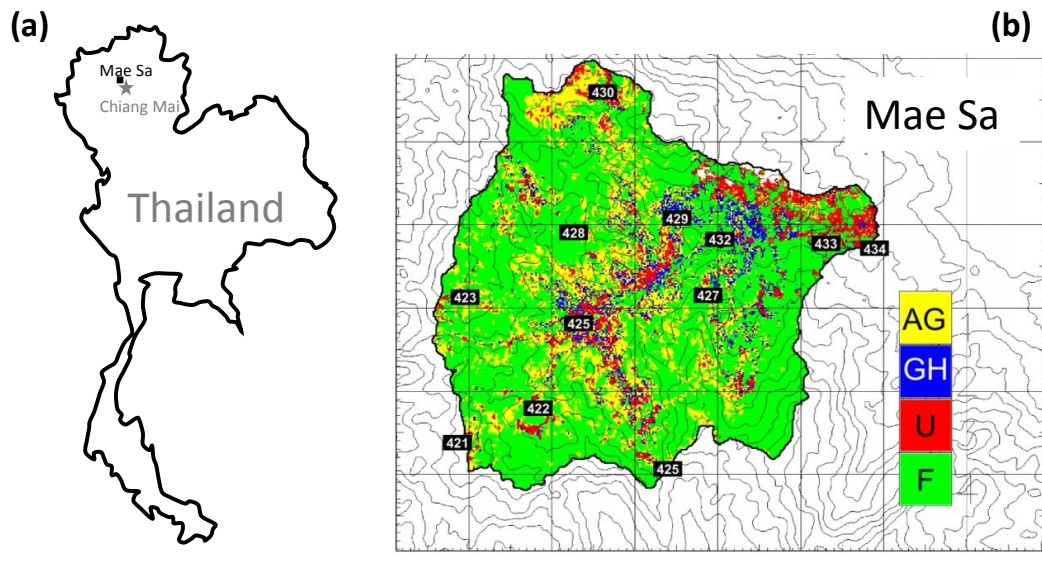





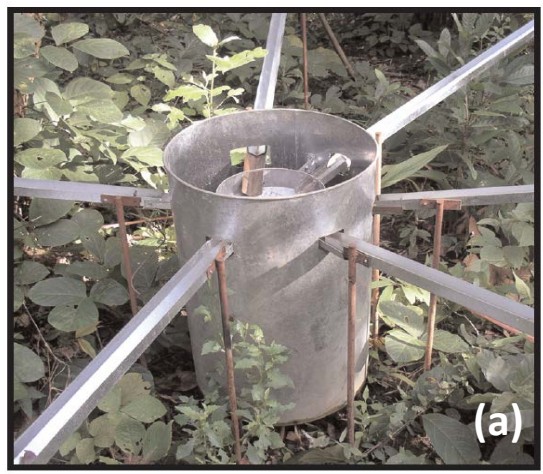

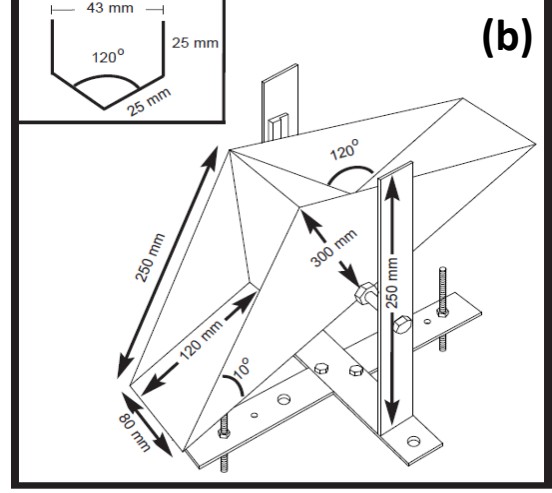





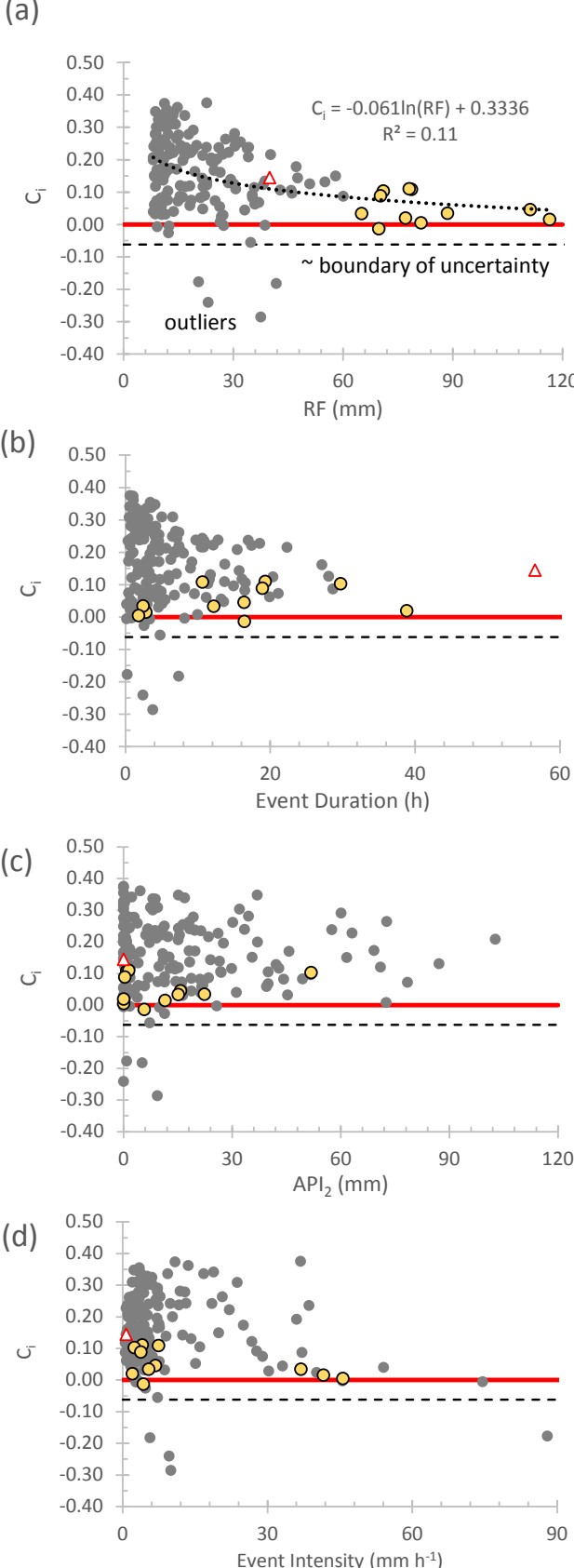

Figure 3




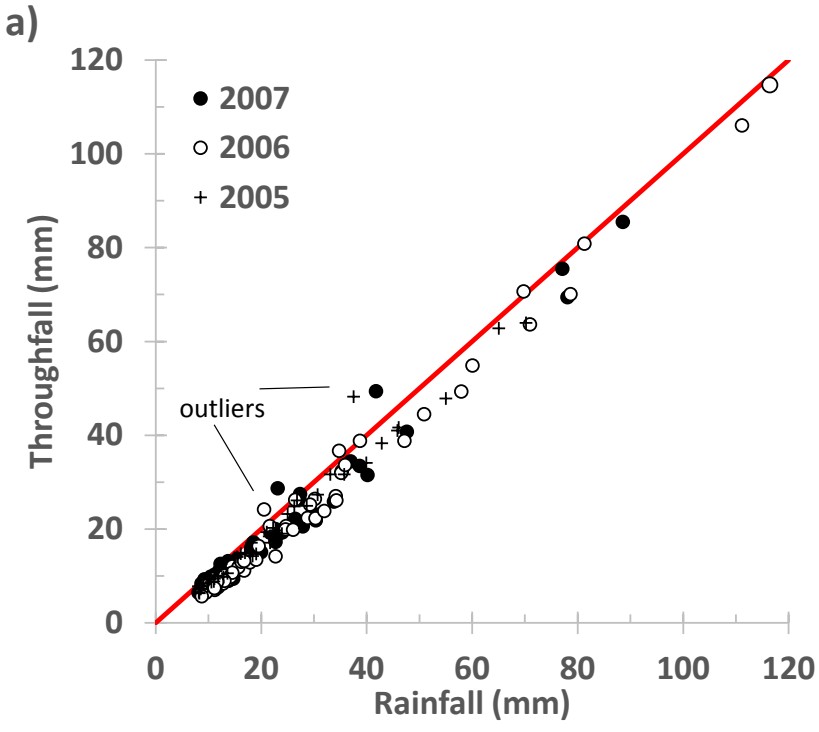

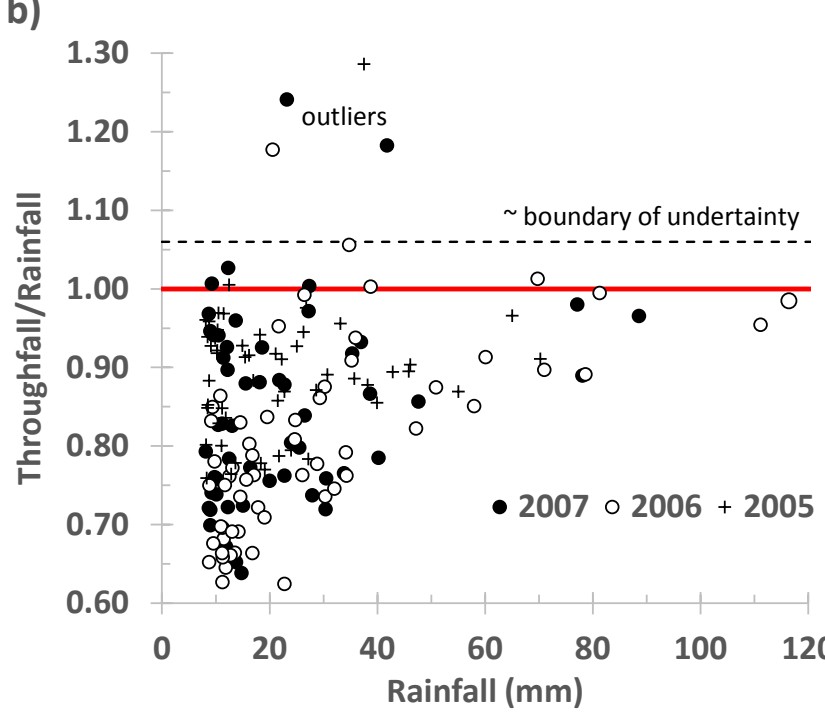

Figure 4



Figure 5





**a) Event #56**

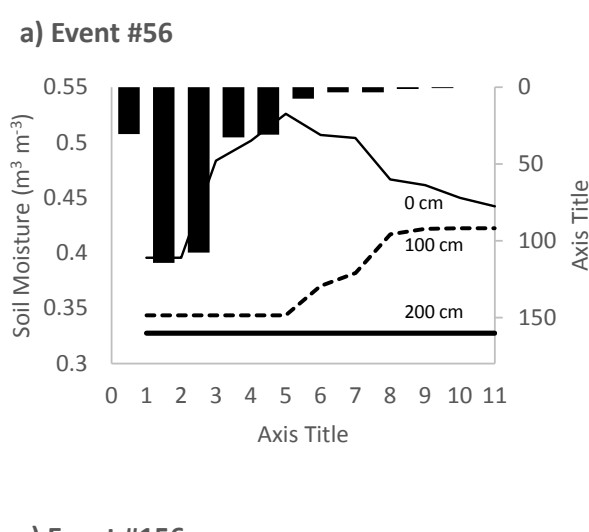

**b) Event #99**

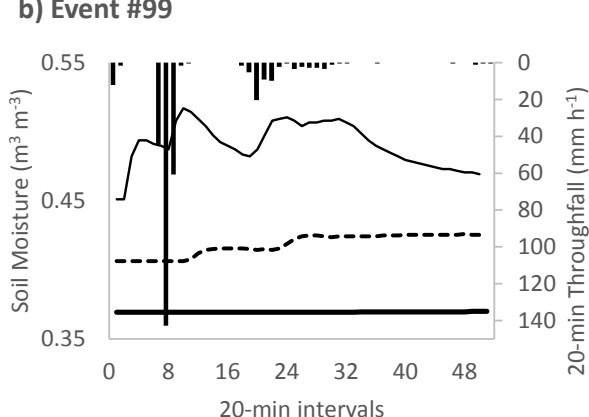

**c) Event #156**

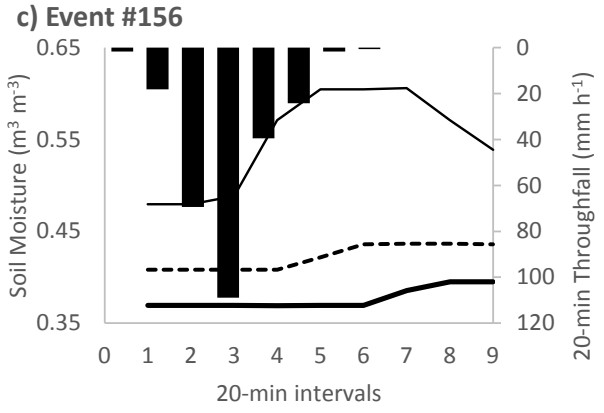

**d) Event #54**

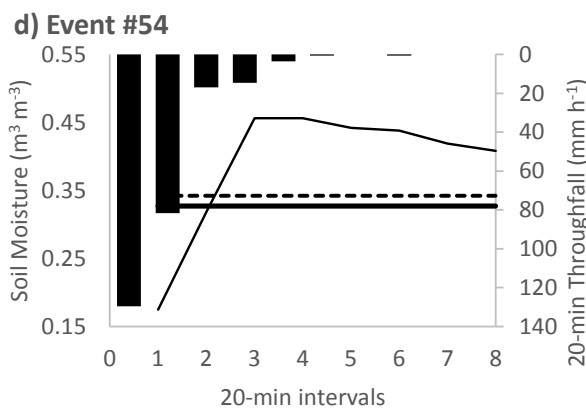

**e) Event #63**

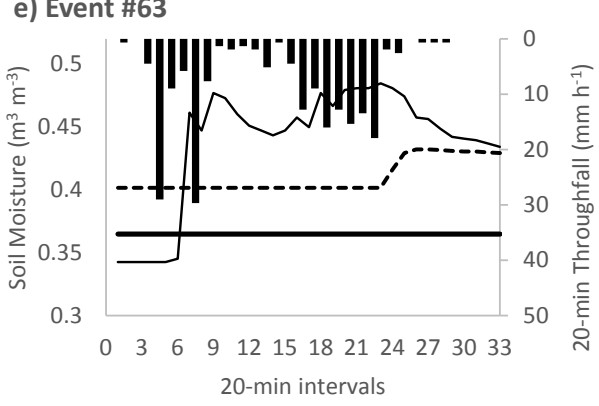

**f) Event #158**

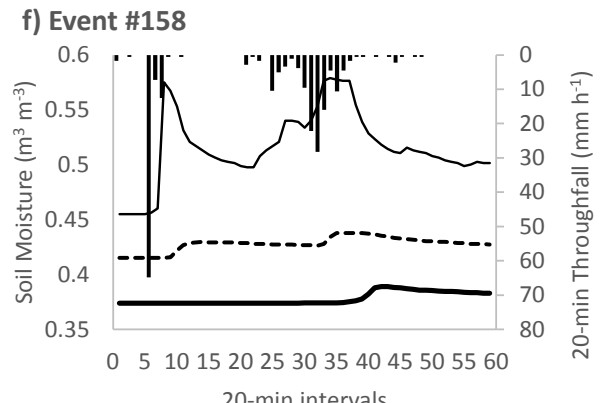

Figure 6



Figure 7