# Peer review of "The canopy interception–landslide initiation conundrum: insight from a tropical secondary forest in northern Thailand"

_Hydrology and Earth System Sciences, 2016_

## Referee Comment (RC1) · Anonymous Referee #1 · 30 May 2016

This manuscript reports on rainfall, throughfall, and soil moisture measurements on a steep slope. The topic is of high relevance to HESS readers and the manuscript is clearly written. It has four important limitations: (1) poor data on throughfall intensity; (2) lack of relevance of soil moisture measurements for slope stability; and (3) overstating conclusions beyond data support:

1. The substantial uncertainty in throughfall measurements using the trough system necessitated large calibration correction of 50-78% (P7L14). The total corrected throughfall estimates are 10-14% less than rainfall, so calibration errors are 3-8 times larger than the difference being quantified. I admire the authors for confessing this limitation, but I don't understand why they find data from this instrument sufficient for

addressing their hypotheses. I also think the calibration procedure cannot be omitted from the paper given its paramount importance. There are some apparent absurdities in the data that may be explainable by the calibration procedure as well: e.g., it appears total TF>RF for the early portions of the most intense storms in the dataset (Fig 5a-d), followed by RF>TF later in the storm during low-intensity RF; I am not aware of this pattern ever being reported. It appears that conditioning the calibration on total TF:RF may have resulted in plausible estimates of total mass balance (where they can be compared to expectations), but measurements of intensity are both the least reliable and most important data. I disagree with the "clarity" claimed P17L22.

2. Hillslope hydrology is poorly constrained, so it is difficult to understand relevance of the soil moisture data to slope stability. There are two problems in the manuscript that arise because of this. (1) The instrumented slope was obviously not near failure during conditions represented in the dataset as evidenced by low soil moisture at depth, and the deep and highly weathered soils suggest this site is not prone to failure in general. It is unclear specifically how soil moisture responses in this slope is useful for understanding slope stability, but the lack of responsiveness at timescales relevant to canopy interception is not enough evidence to conclude a general lack of canopy interception effect. (2) The analysis of paired TF and soil water measurements implies a one-dimensional water balance is relevant for slope failure, but in fact hillslope- and watershed-scale hydrologic conditions are important. Depending on slope configuration, there may be little reason to expect substantial effect of local canopy interception on soil moisture at depth and thus slope stability. These conceptual problems can be addressed by modifying the discussion, but I think the conclusion linking interception to stability through soil moisture at this site (P17L26) is oversimplified.

3a. Intensity-duration quantification of slope stability is useful for general purposes, but limitations of the concept prevent literal application of thresholds. Obviously none of the thresholds were correct for the instrumented slope or it would have failed about 30 times in the 30 months of monitoring. So, each slope must have its own threshold, and
presumably some slopes have thresholds that pass between paired TF and RF intensities in triggering storms (Fig 7). How many slopes is that? The answer to that question is the true effect of canopy interception on slope stability, and the effect of canopy interception on stability of one instrumented slope cannot be reasonably extrapolated to encompass all slopes.

3b. The strong conclusion that there was no intensity smoothing (P15L30) is dubious and contradicted elsewhere in the manuscript (P16L27). In "large events" (Fig 7d) and in most events overall (Fig 7), storm-total TF intensity was lower intensity than RF, so in that sense there was smoothing. Later statements (P16L4-7) rightly focus on peak intensities, but are based on highly uncertain data. Blanket characterization of "no effect" is not credible.

Minor points:

P3L10-18 why present a review of root reinforcement literature when this work has nothing to do with root reinforcement?

P6L23 can you use these field data to convert soil moisture content (m3/m-3) to % saturation? The Results and Discussion refer to degree of saturation (e.g., P13L9, P13L18, P16L16) but no information is presented in the figures or text on porosity or soil moisture release curves and the reader cannot link volumetric soil moisture data to pore pressure.

There are some problems with the figures to clean up. Fig 6a: "axis title"; Fig 6c rainfall bars are not at the same interval as the time interval labels; panels in Fig 5-7 are often different sizes and not aligned.

P14L8 editing error muddles a critical statement about the TF-RF comparison.

---

## Referee Comment (RC2) · Anonymous Referee #2 · 10 Jun 2016

**Reviewer's comment on "The canopy interception-landslide initiation conundrum: insight from a tropical secondary forest in northern Thailand", by R. C. Sidle and A. D. Ziegler**

*General comment*

The manuscript deals with a topic falling within the scope of HESS, to which part of the readership will be interested in. The paper is well structured and clearly written, and the presented experimental data are innovative, as very few examples of similar measurements can be found in the literature. Apart of this merit, however, as the focus of the paper is about the possible effects of canopy interception on the triggering of shallow landslides caused by infiltration into the soil mantle up to a depth of 2 m, the analysis of the results in view of the infiltration processes is poor, lacking important information about soil properties, and more in-depth discussion of the soil moisture dynamics should be provided.

Therefore, my recommendation is that some major revisions are needed before this manuscript could be published in HESS. Some of the following detailed comments will hopefully clarify my point of view.

*Detailed comments*

Page 5, lines 13-14 (minor issue). Please clarify the meaning of "landslides (…) associated with road runoff". A clear definition of the possible triggering mechanisms of landslides in the area would indeed help to better focus the discussion of the measured soil moisture responses to precipitations.

Page 6, lines 6-10 (major issue). Providing more information about soil properties would allow a better understanding of the observed soil moisture changes. Soil porosity is not given, but in the following section 4.4 the authors state that when volumetric moisture content approaches 0.45 the soil is saturated. The provided bulk density data seem to indicate that, at least in the upper layer, the porosity should be greater (by the way, what is the moisture content corresponding to the provided values bulk density?). As the following discussion points out that the triggering of landslide is expected to occur at depths >1.0m, would it be possible to get some information about soil properties (at least porosity and $k_{sat}$) at depths larger than 25cm? (indeed, the authors say that the upper 20cm are characterized by a soil horizon different from the deeper one).

Page 6, lines 15-16 (typo). I think it should read "(stations 429, figures 1b)".

Page 6, line 17 (major issue). The definition of an event should be motivated in view of the expected triggering mechanism. Why the thresholds of 8mm and 4hours have been chosen?

Page 7, lines 6-8 (minor issue). The "dynamic calibration correction" is not clear. Please provide some description of the applied correction.

Page 7, lines 25-30 (moderate issue: I don't know if this issue is minor or major). It is clear that using a large throughfall collector allows the integration over a relatively large area of an inherently inhomogeneous process (in space). However, in the following discussion, in some cases the authors point out that, owing to differences in canopy structure and to the effects of wind (and possibly also to the effects of rainfall intensity, I would add), the dripping of throughfall from canopy could follow different paths, leading to local concentration of drops. How did the authors conclude that the shape, size and position of their collector are adequate? What do the authors think about using several randomly distributed ordinary rain gages? In such a case it could be possible to get information about the adequacy of the obtained spatial mean by

subtracting one (or more) gages and then check if the obtained (spatially averaged) throughfall is affected or not.

Page 9, line 28 – page 10, line 1 (minor issue). The outliers could be an artifact due to concentration of throughfalling drops in the collector, caused by the shape and position of the adopted collector.

Page 11, lines 16-25, and figure 5 (moderate issue). Looking at the provided hyetographs, it seems simply that, regardless of the timing of a peak within the event, when the intensity is below 1.0-1.1 mm/min, it results RF>TF, while it is the other way around when the intensity is larger.

Section 4.4, as a whole (major issue). The whole discussion is too simplistic, and some deeper interpretation should be made. I just give some possible keys. In a soil with $k_{sat}$<5mm/h at the depth of 25cm (and maybe further reducing with depth), it is easily expectable that it may take many hours before water reaches 2.0m depth (even if we don't know soil properties at depth larger than 25cm), so I strongly suggest to extend the time interval over which the soil moisture changes are visualized and discussed (this issue has to do also with the previously raised issue about the adopted definition of a rainfall event). The interpretation of the (clearly visible) effect of initial soil moisture on the effectiveness of a rain event on the following soil moisture changes should be linked to the degree of saturation (but we don't know soil porosity) of the soil and to its hydraulic conductivity (once saturated, the upper layer cannot retain more water, and so, if the hydraulic conductivity allows it, it is "obliged" to release the excess water to the underlying soil). In other words, there should be a maximum storable soil moisture increase, depending on initial moisture condition, over which the excess water penetrates deeper or runs off laterally (above or below surface, or both).

Page 13, lines 5-6 (major issue). It seems to me that limiting the observation of soil moisture to the (widely variable) duration of rainfall events in many cases may be the reason why a (later) deep soil moisture change was not detected.

Page 15, line 22 (typo). It should probably be "environmental conditions  during the storm".

Page 16, lines 16-21 (major issue). See my previous comment about section 4.4. As RF and TF are quite similar in the considered forest, this paragraph would mislead the reader to the conclusion that soil moisture at 2.0m would not be affected by any rainfall event.

Page 16, lines 30-31 (major issue). It is clear that for the considered rain events canopy interception has negligible effects. But, as I already commented above, the rain events have been defined arbitrarily >8,0mm, and there is (maybe obvious) evidence that canopy interception could be larger for smaller events. Could these neglected smaller events affect the initial moisture state of the soil at the beginning of the considered larger events? And, if so, can the authors exclude that canopy interception may play a role in the establishment of such initial moisture state? I would like to read some discussion about this point, before concluding that canopy interception has no effect on landslide initiation.

Page 17, section 6 as a whole (major issue). In view of the previously raised issues, some of the conclusion drawn could be different.

Figure 1, caption (minor issue). It does not seem that the topography and the major stream channels are actually shown in Figure 1a.

Figure 6a (typo). The title of x-axis is missing.

---

## Author Comment (AC1) · 17 Aug 2016

Dear Editor:

We greatly appreciate the efforts of the two reviewers and we have carefully considered and responded to all of their comments with appropriate changes to our manuscript. Our responses are shown here (in blue typeset), including we made our changes and additions. The page and line numbers we report for our changes refer to those in the edited manuscript (marked up) that was submitted. Thank you all for your diligence and we feel that the paper has been greatly improved based on these critiques. We hope that these major revisions will satisfy the reviewers and Editor.

**Anonymous Referee #1**

This manuscript reports on rainfall, throughfall, and soil moisture measurements on a steep slope. The topic is of high relevance to HESS readers and the manuscript is clearly written. It has four important limitations: (1) poor data on throughfall intensity; (2) lack of relevance of soil moisture measurements for slope stability; and (3) overstating conclusions beyond data support:

Thank you for your review comments, these are addressed in the specific responses to your comments that follow.

1. The substantial uncertainty in throughfall measurements using the trough system necessitated large calibration correction of 50-78% (P7L14). The total corrected throughfall estimates are 10-14% less than rainfall, so calibration errors are 3-8 times larger than the difference being quantified. I admire the authors for confessing this limitation, but I don't understand why they find data from this instrument sufficient for addressing their hypotheses. I also think the calibration procedure cannot be omitted from the paper given its paramount importance. There are some apparent absurdities in the data that may be explainable by the calibration procedure as well: e.g., it appears total TF>RF for the early portions of the most intense storms in the dataset (Fig 5a-d), followed by RF>TF later in the storm during low-intensity RF; I am not aware of this pattern ever being reported. It appears that conditioning the calibration on total TF:RF may have resulted in plausible estimates of total mass balance (where they can be compared to expectations), but measurements of intensity are both the least reliable and most important data. I disagree with the "clarity" claimed P17L22.

Thank you for raising this query. After careful consideration, we chose to address this issue in a new section of the paper entitled "Limitations and recommendations":

**6 Limitations and recommendations**

There are some important limitations to our methods. In a prior study, Ziegler et al. (2009) compared the same troughs used herein with several movable tipping bucket gauges, finding no statistical difference between the two approaches. However, in the previous study, total event precipitation was examined, not minutely changes occurring over the course of storms. We caution that the trough method may create a somewhat confusing signal because the area-integrated pattern of throughfall, in which the records are delayed as water captured in the trough flows through the trough, compared to an individual rain gauge placed above the canopy. Nevertheless, at high rates of throughfall, through flow would be more efficient. Additionally, estimates of both throughfall and rainfall have errors. Measurements by tipping-bucket rain gauges installed above a canopy are affected by turbulent exchange at this interface, and wind affects rainfall catch (e.g., Kamph and Burges, 2010). As mentioned before, the

throughfall troughs had a large associated splash error during high-intensity events. While our attempt to correct the splash error resulted in reasonable total event values, individual minutely values could still have substantial errors; these would affect the time series we compare with measured rainfall (Figure 6). However, such limitations are inherent in most all studies reported in the literature (albeit typically not articulated).

In future experiments, we urge researchers to minimize uncertainties by: (a) using troughs that are deeper to minimize splash loss; (b) collect ambient rainfall in more than one location, preferably gauges positioned just above ground level with appropriate wind shields to minimize wind effects; (c) collect more throughfall values, potentially employing other types of collection devices to help interpret the measurements; and (d) preform a rigorous assessment of splash losses to facilitate error correction (again our splash error data are few). We also encourage researchers to spend time in the forest plots during events, recording the various types of phenomena that may affect the capture of throughfall over time, to help with the interpretation of data. Here, we re-emphasize that throughfall reaching the forest floor is highly variable in space and time. Multi-stored canopies can create wet and dry zones below them, which change over the course of a storm with respect to variable wind direction, changes in rainfall phenomena (rate, drop size), and changes in canopy wetness (e.g., Konishi et al., 2006). The oddities apparent in the data of some of our recorded storms (e.g., higher throughfall than rainfall during some periods, but not others) may possibly be related to inadequacies in our error correction; however, these may be realistic. For example, they may result because wind-driven rain is captured by portions of a large tree and then channelled directly to the throughfall trough (as was documented at the prior study site; Ziegler et al., 2009).

We believe that these uncertainties do not undermine the integrity of our conclusions. While the uncertainties may prevent us from producing a high-precision budget of the portion of rainfall converted to throughfall at minutely scales, they do allow us to address the primary goals of this investigation, which are to assess whether secondary tropical canopies intercept sufficient rainwater during large storms to mitigate landslide initiation compared to open areas.

2. Hillslope hydrology is poorly constrained, so it is difficult to understand relevance of the soil moisture data to slope stability. There are two problems in the manuscript that arise because of this. (1) The instrumented slope was obviously not near failure during conditions represented in the dataset as evidenced by low soil moisture at depth, and the deep and highly weathered soils suggest this site is not prone to failure in general. It is unclear specifically how soil moisture responses in this slope is useful for understanding slope stability, but the lack of responsiveness at timescales relevant to canopy interception is not enough evidence to conclude a general lack of canopy interception effect. (2) The analysis of paired TF and soil water measurements implies a one-dimensional water balance is relevant for slope failure, but in fact hillslope- and watershed-scale hydrologic conditions are important. Depending on slope configuration, there may be little reason to expect substantial effect of local canopy interception on soil moisture at depth and thus slope stability. These conceptual problems can be addressed by modifying the discussion, but I think the conclusion linking interception to stability through soil moisture at this site (P17L26) is oversimplified.

While we agree that the slope hydrology (actually the soil physics) could have been better constrained, we emphasize this was only a supplemental part of the study – i.e., as stated in the third objective "(to) determine the effect of canopy interception on the potential for soil water increases that could trigger landslides". True, the site was not near failure during the storms that were monitored, in spite of some of them exceeding the conservative global slope failure thresholds for intensity – duration. It is also true that for our site we were only able to assess one-dimensional (vertical) flow. But we never intended this to be an assessment of stability of

3-D hillslopes where issues like convergent topography and related flow pathways exert controls on landslide initiation. We simply wanted to address the issue of the role of canopy interception during storms of different sizes on the delivery of rainwater to and into the soil. Our inclusion of the soil moisture data was simply to show that rather homogeneous soils (without extensive macropores) could effectively buffer peak rainfall inputs that have been postulated by others to affect pore water pressure at depth and thus affect landslides. We recognize the limitations of such a 1-D assumption in the context of 3-D hillslope hydrology and now address this more clearly. The reviewer is correct in stating that "Depending on slope configuration, there may be little reason to expect substantial effect of local canopy interception on soil moisture at depth and thus slope stability.", and we have modified the Discussion to note this more clearly. But this is clearly not recognized by a number of researchers who keep supporting the idea that canopy interception buffers rain inputs and potentially ameliorates peak pore water pressure response at depth. Additionally, to address these concerns were have modified the "Conclusions".

See additions on pg. 12, L 20-23 and pg. 17, L 13.

3a. Intensity-duration quantification of slope stability is useful for general purposes, but limitations of the concept prevent literal application of thresholds. Obviously none of the thresholds were correct for the instrumented slope or it would have failed about 30 times in the 30 months of monitoring. So, each slope must have its own threshold, and presumably some slopes have thresholds that pass between paired TF and RF intensities in triggering storms (Fig 7). How many slopes is that? The answer to that question is the true effect of canopy interception on slope stability, and the effect of canopy interception on stability of one instrumented slope cannot be reasonably extrapolated to encompass all slopes.

Please understand that the intensity – duration 'thresholds' are the minimum combinations of average storm intensity and duration which have been recorded that have triggered landslides somewhere in the world. Exceedance of thresholds does not mean a landslide will occur, it means that the very minimum rainfall conditions for global landslides has been met. True, it is a very general indicator of landslide susceptibility, but in countries like Thailand where detailed geotechnical, soils, and geological measurements are not widely available, such simple rainfall – landslide relations may be useful. And, of course, we agree that each hillslope (even hillslope segment) has a unique threshold for actual slope failure based on a number of factors and predispositions, but the point here was to show that canopy interception during storms that exceeded the global thresholds did not have a big influence on rainwater delivery to the soil surface. The question about how many hillslopes 'have unique thresholds that pass between the throughfall and rainfall intensities' seems unclear and rhetorical, but it was never our intention to apply our findings to the catchment-scale, rather we aimed to clarify the effects of canopy interception on water delivery to the soil surface (where much speculation occurs). We have noted the reviewer's concerns and have tried to address these by additions on pg. 8, L. 16-17 and pg. 9, L. 5-7, plus other places.

3b. The strong conclusion that there was no intensity smoothing (P15L30) is dubious and contradicted elsewhere in the manuscript (P16L27). In "large events" (Fig 7d) and in most events overall (Fig 7), storm-total TF intensity was lower intensity than RF, so in that sense there was smoothing. Later statements (P16L4-7) rightly focus on peak intensities, but are based on highly uncertain data. Blanket characterization of "no effect" is not credible.

The statement beginning on Pg. 16, L27 does not state or infer that there was intensity smoothing by the canopy. Smoothing of rainfall peaks cannot be derived from Figure 7 – these intensity values are conservative, average storm intensity (this was originally stated on pg. 8, L 14 and on pg. 18, L 1, and is now emphasized on pg. 8, L 16, pg. 9 , L5-7; pg. 14, L 11 and 27; Pg. 17, L 21). The fact that average intensity over the course of an entire storm is less in TF compared to rainfall does not necessarily imply smoothing of rainfall peaks. Nevertheless, we have modified our statement in the Discussion (see pg. 16, L 21-22, 29).

Minor points:

P3L10-18 why present a review of root reinforcement literature when this work has nothing to do with root reinforcement?

We have now reduced this information on root reinforcement. We feel it serves the purpose of putting the issues of vegetation effects on slope stability into perspective, but, as suggested, we have reduced the reference to root reinforcement to one sentence. See changes to second paragraph of the Introduction.

P6L23 can you use these field data to convert soil moisture content (m3/m-3) to %saturation? The Results and Discussion refer to degree of saturation (e.g., P13L9, P13L18, P16L16) but no information is presented in the figures or text on porosity or soil moisture release curves and the reader cannot link volumetric soil moisture data to pore pressure.

This information is now presented in the text, see query below. In addition, in Table 2 we now indicate the maximum wetness (relative to saturation) that occurs during an event, extending to 24-h after the event ends.

There are some problems with the figures to clean up. Fig 6a: "axis title"; Fig 6c rainfall bars are not at the same interval as the time interval labels; panels in Fig 5-7 are often different sizes and not aligned.

All the figures 5-7 have been redrawn, with special attention paid to alignment of the axis.

P14L8 editing error muddies a critical statement about the TF-RF comparison.

The sentence "The three events in which incident rainfall exceeded the threshold but throughfall did not were similar in canopy interception ($C_i$ = 0.20-0.23); duration (3.4-4.6 h); and event intensity (7.1- 7.7 mm h-1)".

Was modified to:

"The three events in which incident rainfall exceeded the threshold, but throughfall did not, were similar in canopy interception ($C_i$ = 0.20-0.23); duration (3.4-4.6 h); and event intensity (7.1- 7.7 mm $h^{-1}$)".

**Anonymous Reviewer #2**

*General comment*
The manuscript deals with a topic falling within the scope of HESS, to which part of the readership will be interested in. The paper is well structured and clearly written, and the

presented experimental data are innovative, as very few examples of similar measurements can be found in the literature. Apart of this merit, however, as the focus of the paper is about the possible effects of canopy interception on the triggering of shallow landslides caused by infiltration into the soil mantle up to a depth of 2 m, the analysis of the results in view of the infiltration processes is poor, lacking important information about soil properties, and more in-depth discussion of the soil moisture dynamics should be provided.

Thank you for recognizing the uniqueness of this research. Actually the focus of the paper is not on the effects of infiltration processes into the soil mantle; as stated in the third objective "(to) determine the effect of canopy interception on the potential for soil water increases that could trigger landslides", this was only a supplemental part of the study. We have clarified that our data only assessed one-dimensional (vertical) soil moisture fluxes (pg. 12, L 20-22) and that we did not attempt to assess the effect of slope shape on subsurface water flux (pg. 17, L 14). Our inclusion of the soil moisture data was simply to show that homogeneous soils could buffer peak rainfall inputs (with or without canopy interception) in contrast to other speculations that canopies alone can buffer pore water pressure at depth and thus affect landslides. Our primary objective was to assess the role, magnitude, and timing of tropical canopy interception during storms of different sizes on the delivery of rainwater to and into the soil. Please see our responses that follow to specific comments.

Therefore, my recommendation is that some major revisions are needed before this manuscript could be published in HESS. Some of the following detailed comments will hopefully clarify my point of view.

*Detailed comments*
Page 5, lines 13-14 (minor issue). Please clarify the meaning of "landslides (…) associated with road runoff". A clear definition of the possible triggering mechanisms of landslides in the area would indeed help to better focus the discussion of the measured soil moisture responses to precipitations.

We have added text and a reference to clarify the landslide triggering mechanisms; pg. 5, L14-16.

Page 6, lines 6-10 (major issue). Providing more information about soil properties would allow a better understanding of the observed soil moisture changes. Soil porosity is not given, but in the following section 4.4 the authors state that when volumetric moisture content approaches 0.45 the soil is saturated. The provided bulk density data seem to indicate that, at least in the upper layer, the porosity should be greater (by the way, what is the moisture content corresponding to the provided values bulk density?). As the following discussion points out that the triggering of landslide is expected to occur at depths >1.0m, would it be possible to get some information about soil properties (at least porosity and ksat) at depths larger than 25cm? (indeed, the authors say that the upper 20cm are characterized by a soil horizon different from the deeper one).

We have added information on soil properties, particularly, bulk density, porosity, and saturated hydraulic conductivity as follows (pg. 6, L 9-17):

> The soil at the site is an Ultisol with a thin ($< 20$ cm) brown A horizon underlain by a dark red B horizon that extends below a depth of two meters. Saturated hydraulic conductivity declines exponentially from the surface (~136 mm h$^{-1}$) to approximately

< 4 mm h$^{-1}$ at a depth of 25 cm; values at 1 m and 2 m are 1-2 mm h$^{-1}$ (n = 3 measurements for all depths; unpublished data, determined with a bore-hole permeameter). Bulk density does not change much over this depth range (1.08-1.38 g cm$^{-3}$ for the surface, 1 m. and 2 m depths). Corresponding porosity for the three depths is 0.59, 0.52, and 0.48 (based on a particle density of 2.65 g cm$^{-3}$). The decrease in saturated hydraulic conductivity is typical of that in other profiles found in southeast Asia (cf. Ziegler et al., 2004; 2006). Macropores and fissures, features that could influence preferential flow through the soil, were not abundant in the subsoil.

During the storms we did not observe the occurrence of a perched water table, which implies that there is no restricting K layer at depths (at least down to 2 m). Saturation was only approached at the surface for a brief period of time (see Table 2 and Figure 6). Bulk density is reported for dry soil conditions as typical in the soils literature.

Page 6, lines 15-16 (typo). I think it should read "(stations 429, figures 1b)".

Changed to "station 429, Figure 1b"

Page 6, line 17 (major issue). The definition of an event should be motivated in view of the expected triggering mechanism. Why the thresholds of 8mm and 4hours have been chosen?

Our objective was to examine all rainstorms that occurred during this ≈ 3 yr period. We wanted to show how interception varies for different size events. Later in the paper we address how bigger storms (e.g., Table 2, Figures 5 and 6 and related discussions), which are of the size that could potentially trigger landslides, affect interception and water delivery to and into the soil. Some clarification is now included on page 6, L 24-25: "As such, we included a range of monsoon storms to assess interception losses for potentially landslide-triggering events and those that were smaller."

Page 7, lines 6-8 (minor issue). The "dynamic calibration correction" is not clear. Please provide some description of the applied correction.

The following was modified/added (with new references):

A dynamic calibration correction was then applied to account for differences in tip volume over the range of observed tipping rates (Ziegler et al., 2009). Dynamic calibration accounts for differences in tip volume over the range of observed tipping rates (Calder and Kidd, 1978; Marsalek, 1981; Humphrey et al.,1997). These differences are caused by "splash" losses occurring as the tipping mechanism is moving when rainfall is draining from the funnel. This relationship was determined by draining known volumes of water through the tipping bucket system (mm tip$^{-1}$) and recording the number of tips registered.

Page 7, lines 25-30 (moderate issue: I don't know if this issue is minor or major). It is clear that using a large throughfall collector allows the integration over a relatively large area of an inherently inhomogeneous process (in space). However, in the following discussion, in some cases the authors point out that, owing to differences in canopy structure and to the effects of wind (and possibly also to the effects of rainfall intensity, I would add), the dripping of throughfall from canopy could follow different paths, leading to local concentration of drops.

How did the authors conclude that the shape, size and position of their collector are adequate? What do the authors think about using several randomly distributed ordinary rain gages? In such a case it could be possible to get information about the adequacy of the obtained spatial mean by subtracting one (or more) gages and then check if the obtained (spatially averaged) throughfall is affected or not.

In a prior work we compared the same troughs used herein with several movable tipping-bucket (round) gauges. In that work we found no statistical difference between the two. However, we were looking at "event" totals, not dynamic changes over the course of a storm. It might be that the trough method leads to a confusing signal because the integrated pattern of throughfall, which has a delay as the water flows down the trough, is being compared to an individual rain gauge placed above the canopy. In this paper we now speak of this potential problem in the new limitations section, which is introduced above in the Reviewer 1 queries.

Page 9, line 28 – page 10, line 1 (minor issue). The outliers could be an artifact due to concentration of throughfalling drops in the collector, caused by the shape and position of the adopted collector.

This comment cuts to the issue of using a combined "Results and Discussion" rather than separating these. In our original submission, we combined these sections, but were then asked to separate them. The answer to your question appears on pg. 15, L 29-31, but we now made changes to the last sentence of this paragraph which appears on pg. 10, L 9-11.

Page 11, lines 16-25, and figure 5 (moderate issue). Looking at the provided hyetographs, it seems simply that, regardless of the timing of a peak within the event, when the intensity is below 1.0-1.1 mm/min, it results RF>TF, while it is the other way around when the intensity is larger.

You make a good point, but this is not true for event #63 (and some other events as well that are not shown in Fig. 5). To clarify, we added the following sentence on pg. 12, L13-15: While in five of these six large events rainfall exceeded throughfall when intensities were $< 1.0 – 1.1$ mm min$^{-1}$ and, typically, throughfall exceeded rainfall when intensities were $> 1.1$ mm min$^{-1}$, this pattern was not consistently found in other storms.

Section 4.4, as a whole (major issue). The whole discussion is too simplistic, and some deeper interpretation should be made. I just give some possible keys. In a soil with ksat<5mm/h at the depth of 25cm (and maybe further reducing with depth), it is easily expectable that it may take many hours before water reaches 2.0m depth (even if we don't know soil properties at depth larger than 25cm), so I strongly suggest to extend the time interval over which the soil moisture changes are visualized and discussed (this issue has to do also with the previously raised issue about the adopted definition of a rainfall event). The interpretation of the (clearly visible) effect of initial soil moisture on the effectiveness of a rain event on the following soil moisture changes should be linked to the degree of saturation (but we don't know soil porosity) of the soil and to its hydraulic conductivity (once saturated, the upper layer cannot retain more water, and so, if the hydraulic conductivity allows it, it is "obliged" to release the excess water to the underlying soil). In other words, there should be a maximum storable soil moisture increase, depending on initial moisture condition, over which the excess water penetrates deeper or runs off laterally (above or below surface, or both).

We examined the soil moisture response at depth (1 m and 2 m) for all events and these longer term data are now presented in Fig. 6 – i.e., we have extended the time interval on the x-axis by 2- to 10-fold in order to show where the small soil moisture increases at depths 1 and 2 m completely subside. The degree of saturation is now show in three new columns (Wetness, 0 m, 1 m, and 2 m) introduced in the revised Table 2. Please note that during only two events (#156 and 158) was saturation reached (or nearly reached) in the surface soil; and in no events was the subsoil approaching saturation. Please see rather extensive changes and additions have been made based on these longer time assessments referenced to new data now shown in Fig. 6 and Table 2 (see pg. 13, L 17 to pg. 14, L8); however, please note that these changes do not affect our general conclusions.

Page 13, lines 5-6 (major issue). It seems to me that limiting the observation of soil moisture to the (widely variable) duration of rainfall events in many cases may be the reason why a (later) deep soil moisture change was not detected.

See response to previous comment and changes made on pg. 13, L 17 to pg. 14, L8.

Page 15, line 22 (typo). It should probably be "environmental conditions change during the storm".

Corrected (now pg. 16, L 13)

Page 16, lines 16-21 (major issue). See my previous comment about section 4.4. As RF and TF are quite similar in the considered forest, this paragraph would mislead the reader to the conclusion that soil moisture at 2.0m would not be affected by any rainfall event.

We have addressed this issue in our revisions in section 4.4, and we also have modified the text accordingly in the Discussion section as requested: see pg. 17, L 8-15.

Page 16, lines 30-31 (major issue). It is clear that for the considered rain events canopy interception has negligible effects. But, as I already commented above, the rain events have been defined arbitrarily >8,0mm, and there is (maybe obvious) evidence that canopy interception could be larger for smaller events. Could these neglected smaller events affect the initial moisture state of the soil at the beginning of the considered larger events? And, if so, can the authors exclude that canopy interception may play a role in the establishment of such initial moisture state? I would like to read some discussion about this point, before concluding that canopy interception has no effect on landslide initiation.

We agree. Certainly interception versus rainfall differences for very small events that just precede a large storm could affect soil moisture response. However, combined evapotranspiration and infiltration would negate such effects after a day or at most a couple days (depending on weather conditions). We have added a comment to address this issue on pg. 17, L 13-15.

Page 17, section 6 as a whole (major issue). In view of the previously raised issues, some of the conclusion drawn could be different.

We have modified our "Conclusions" section accordingly.

Figure 1, caption (minor issue). It does not seem that the topography and the major stream channels are actually shown in Figure 1a.

The figure has been revised to show this information:

> Figure 1: Site map of the Mae Sa experiment site in northern Thailand. Panel (a) shows the catchment location in Thailand, major contours, and the stream system. Panel (b) shows the major land covers in the Mae Sa catchment including hillslope and plantation agriculture (AG, 23%), greenhouse agriculture (GH, 7%), urbanized or peri-urban areas (U, 8%), and forest cover with various degrees of disturbance (F, 62%). Grid cell dimensions are 2 x 2 km. Rectangles demarcate hydro-meteorological measurement sites. Streamflow, total suspended solids, particulate organic carbon, and particulate organic nitrogen were measured at the stream gage station 434. Rainfall is measured at all other numbered hydro-meteorological stations (rectangles). The throughfall investigation in this paper was conducted at station 429, where rainfall, throughfall, and soil moisture where monitored.

Figure 6a (typo). The title of x-axis is missing.

This error has been fixed.

---

## Referee Report (RR1)

**Review of `The canopy interception-landslide initiation conundrum: Insight from a tropical secondary forest in northern Thailand'**

The authors collected and analyzed rainfall interception and soil moisture data at their study site in northern Thailand over a three year period. They examine trends between rainfall intensity, rainfall duration, antecedent precipitation, and throughfall and discuss implications for the effects of canopy interception on the potential for shallow landslide initiation. I think that this manuscript will be of interest to readers of Hydrology and Earth System Sciences. The problem is well motivated and the results generally support the conclusions. The authors have addressed many of the issues raised in previous reviews. In general, I have few substantive comments but think that a few clarifications and presentation of additional data (as noted below) would improve the manuscript.

General Comment: Goal number one of the study is to evaluate rainfall interception by a secondary tropical forest canopy. A large amount of data is presented and I think it would improve the impact of the study if it were possible to synthesize some of it in a few simple ways. For example, would it be possible to determine a canopy storage capacity for this setting from the data?

Section 3.1, Line 25: Explicitly state the motivation for using these rainfall-intensity duration thresholds. I'm guessing it is helpful to roughly identify the types of storms that have the potential to trigger landslides since the effectiveness of interception is known to vary with rainfall intensity and duration.

Page 11, Line 1: There is a lot of focus on antecedent rainfall, but other meteorologic factors may be equally important. For instance, how much does canopy interception at this site depend on the evaporation rate? I imagine this could be critical, especially for the longer events. Evaporation rate could potentially be estimated using data from the hydrometeorological station data.

Page 11, Line 23: Suggest starting a new paragraph with the sentence beginning `The three largest events….'

Page 12, Line 3: `While in five of these six….' Some supporting data showing early storage during other storm events needs to accompany this statement.

Page 16, Line 26: The potential failure plane could be close to the surface in some environments. I suggest making this statement more specific to your field site.

Page 18 Line 23 - Page 19 Line 2: Much of this was stated earlier and could be summarized more succinctly so that the focus remains on the new insights gained in this study.

Page 18, Line 6: Potential failure planes could be very shallow in some settings.

Figure 6: Since bulk density varies with depth, I would suggest plotting percent saturation on the y-axis.

---

## Editor Decision (ED1)

[revised manuscript text omitted]

Figure 7

---

## Author Response (AR2)

**Please see below how we handled the queries from the reviewer with minor concerns. The second reviewer indicated "accept as is".**

**Review of `The canopy interception-landslide initiation conundrum: Insight from a tropical secondary forest in northern Thailand'**

The authors collected and analyzed rainfall interception and soil moisture data at their study site in northern Thailand over a three year period. They examine trends between rainfall intensity, rainfall duration, antecedent precipitation, and throughfall and discuss implications for the effects of canopy interception on the potential for shallow landslide initiation. I think that this manuscript will be of interest to readers of Hydrology and Earth System Sciences. The problem is well motivated and the results generally support the conclusions. The authors have addressed many of the issues raised in previous reviews. In general, I have few substantive comments but think that a few clarifications and presentation of additional data (as noted below) would improve the manuscript.

General Comment: Goal number one of the study is to evaluate rainfall interception by a secondary tropical forest canopy. A large amount of data is presented and I think it would improve the impact of the study if it were possible to synthesize some of it in a few simple ways. For example, would it be possible to determine a canopy storage capacity for this setting from the data?

We added the following sentence in the Discussion on pg. 14 to address this question (and referred to data in Fig. 4b): "*Although the canopy storage capacity is quite variable, especially during small events, based on data in Figure 4b, the upper limits of canopy storage appear to be about 35% of rainfall for small events increasing to nearly no storage in the largest events.*"

Section 3.1, Line 25: Explicitly state the motivation for using these rainfall-intensity duration thresholds. I'm guessing it is helpful to roughly identify the types of storms that have the potential to trigger landslides since the effectiveness of interception is known to vary with rainfall intensity and duration.

At the bottom of pg. 8 we have added the following sentence to clarify: "*Herein we employ these thresholds to ascertain whether interception has a significant effect on the intensity-duration relations that may trigger landslides at our site.*"

Page 11, Line 1: There is a lot of focus on antecedent rainfall, but other meteorologic factors may be equally important. For instance, how much does canopy interception at this site depend on the evaporation rate? I imagine this could be critical, especially for the longer events. Evaporation rate could potentially be estimated using data from the hydrometeorological station data.

In the paper, we mostly use antecedent rainfall because this parameter (API) has been correlated with maximum piezometric response in unstable hollows in other studies and has also been used to segregate 'wet' (API . 20 mm) and 'dry' (API$_2$ ≤ 20 mm) antecedent conditions using intensity–duration relationships.

We agree that other meteorologic factors are important. We report wind conditions in partial effort to address this. An estimate of canopy drying from evaporation is a relevant process, however, we don't have access to all of the data needed to do this, hence we were forced to rely on wind and API data. Considering the variation in Ci values we observed, it is likely that a more accurate measure of canopy drying (stemming from the evaporation estimate), will not affect our results, because the overwhelming variables affecting Ci are rainfall intensity and a rainfall depth that far exceeds canopy storage—both are associated with large storms. It is possible, that drying could reduce throughfall during some of the longer storms with intermittent rainfall, (see in Figure 4 c), but these events are few in the data set. In the absence of the evaporation estimate, we have done the best we could do in terms of mentioning the effect of canopy storage/wetting/drying.

Page 11, Line 23: Suggest starting a new paragraph with the sentence beginning `The three largest events….'

Revised as suggested.

Page 12, Line 3: `While in five of these six….' Some supporting data showing early storage during other storm events needs to accompany this statement.

We agree with the comment. We now refer to Figure 5, where this pattern is shown. The following passage is relevant:

> While in five of these six large events rainfall exceeded throughfall when intensities were < 1.0 – 1.1 mm min$^{-1}$ and, typically, throughfall exceeded rainfall when intensities were > 1.1 mm min$^{-1}$, this pattern was not consistently found in other storms (Figure 5).

Page 16, Line 26: The potential failure plane could be close to the surface in some environments. I suggest making this statement more specific to your field site.

Actually the potential failure plane (for relatively shallow landslides) is typically 2 m or greater in this region. To clarify, we modified the text on pg. 16 as follows: *"Because of the absence of a constricting permeability layer at shallow depths in these deeply weathered soils, most potential failure planes occur at depths of 2 m or greater. At the depth of 2 m only very minor increases in soil moisture ($\leq 0.01$-$0.02$ $m^3$ $m^3$) were recorded …"*

Page 18 Line 23 - Page 19 Line 2: Much of this was stated earlier and could be summarized more succinctly so that the focus remains on the new insights gained in this study.

We agree, the last two paragraphs of the Conclusions have been condensed as follows: *"Few studies have reported intra-storm comparisons of incident rainfall and throughfall at temporal resolutions that could be used to assess effects on shallow landslide initiation (i.e. $\leq 1$ h). While many of these investigations note smoothing effects of canopy interception on incident rain intensity, none show any physical evidence that canopy smoothing lowered soil moisture or pore pressures at depths that would reduce landslide susceptibility. Although our throughfall results from many large and intense monsoon events in northern Thailand were affected by instrumental errors (common in all studies of this type), our results indicate that these secondary tropical forest canopies have relatively small smoothing effects on incident rainfall peaks. We also show that soil moisture response is quite dampened or even non-responsive at depths where potential failure planes exist in this region ($\geq 2$ m). These data coupled with our analysis of mean rain intensity – duration thresholds…"*

Page 18, Line 6: Potential failure planes could be very shallow in some settings.

This issue about potential failure plane depth has now been clarified on pg. 16, L 25-27 and pg. 19 L 1.

Figure 6: Since bulk density varies with depth, I would suggest plotting percent saturation on the y-axis.

This information is now provided as "Wetness" reported in Table 2. The readers can easily see that for all events, the maximum soil moisture is well below saturation for depths of 1m and 2m.

---

## Author Response (AR3)

[revised manuscript text omitted]

**Commented [RS2]:** We changed this to reflect all precipitation during the study period (not just storms), thus the percentage is lower (but still substantial)

[revised manuscript text omitted]